# Polycomb complexes associate with enhancers and promote oncogenic transcriptional programs in cancer through multiple mechanisms

Ho Lam Chan [1,2], Felipe Beckedorff [1,2], Yusheng Zhang[1,2], Jenaro Garcia-Huidobro[1,2,5], Hua Jiang[3], Antonio Colaprico[1,2], Daniel Bilbao[1], Maria E. Figueroa[1,2], John LaCava[3,4], Ramin Shiekhattar[1,2] & Lluis Morey [1,2]

Polycomb repressive complex 1 (PRC1) plays essential roles in cell fate decisions and development. However, its role in cancer is less well understood. Here, we show that *RNF2*, encoding RING1B, and canonical PRC1 (cPRC1) genes are overexpressed in breast cancer. We find that cPRC1 complexes functionally associate with ERα and its pioneer factor FOXA1 in ER+ breast cancer cells, and with BRD4 in triple-negative breast cancer cells (TNBC). While cPRC1 still exerts its repressive function, it is also recruited to oncogenic active enhancers. RING1B regulates enhancer activity and gene transcription not only by promoting the expression of oncogenes but also by regulating chromatin accessibility. Functionally, RING1B plays a divergent role in ER+ and TNBC metastasis. Finally, we show that concomitant recruitment of RING1B to active enhancers occurs across multiple cancers, highlighting an under-explored function of cPRC1 in regulating oncogenic transcriptional programs in cancer.

[1] Sylvester Comprehensive Cancer Center, Biomedical Research Building, 1501 NW 10th Avenue, Miami, FL 33136, USA. [2] Department of Human Genetics, University of Miami Miller School of Medicine, Miami, FL 33136, USA. [3] Laboratory of Cellular and Structural Biology, The Rockefeller University, New York, NY 10065, USA. [4] Institute for Systems Genetics and Department of Biochemistry and Molecular Pharmacology, New York University School of Medicine, New York, NY 10016, USA. [5] Centro de Investigaciones Médicas (CIM), Núcleo Científico Multidisciplinario, Escuela de Medicina, Universidad de Talca, Avenida Lircay S/N, Talca 3460000, Chile. Correspondence and requests for materials should be addressed to L.M. (email: lmorey@med.miami.edu)

Polycomb group genes (PcG) are evolutionarily conserved epigenetic regulators that can be divided into two main complexes, Polycomb repressive complex 1 and 2 (PRC1 and PRC2)[1]. In mammals, the core PRC2 complex contains SUZ12, EED, and the histone methyltransferase enzymes EZH1/2, which catalyze di- and trimethylation on lysine 27 of histone H3 (H3K27me2/3)[2]. The two main PRC1 sub-complexes are the canonical and non-canonical PRC1 complexes (cPRC1 and ncPRC1). cPRC1 comprises PCGF2/4, Polyhomeiotic (PHC1/2/3), the CBX proteins (CBX2/4/6/7/8), and the E3-ligase subunits RING1A/B, which monoubiquitinate histone H2A at lysine 119 (H2AK119ub1). In contrast, ncPRC1 complexes include RYBP/YAF2, PCGF1/3/5, and RING1A/B, as well as additional co-factors[3]. We and others have shown that cPRC1, ncPRC1, and PRC2 complexes regulate stem cell pluripotency, cell fate decisions, and development[4,5]. Historically, Polycomb complexes have been mostly associated with maintaining gene repression. However, increasing evidence indicates that specific PRC1 variants can be recruited to actively transcribed genes in multiple biological processes[6–10].

While PRC1 genes are not typically mutated, they are dysregulated in many cancer types. *BMI1*, encoding for PCGF4, is the best studied PRC1 gene in cancer to date. It is often overexpressed in cancer and is important for tumor initiation and progression[11,12]. In contrast, *PCGF2* is downregulated in prostate and colorectal cancers[13], suggesting that PCGF paralogs have distinct functions in cancer. Recent studies suggested that PRC1 genes that play important roles in cancer carry out their functions independently of their association with PRC1[14,15]. Nonetheless, despite great efforts to understand the epigenetic mechanisms that contribute to human maladies, a comprehensive analysis of genomic alterations of PRC1 genes, and the architecture, function, and activity of PRC1 complexes in cancer, have yet to be fully addressed.

Here, we show that PRC1 genes are genetically amplified in breast cancer. In contrast to its canonical function, RING1B (encoded by *RNF2*) is predominantly recruited to enhancers and specifically regulates oncogenic transcriptional programs in different breast cancer subtypes. Mechanistically, RING1B associates with specific cPRC1 components that are recruited to enhancers containing estrogen receptor alpha (ERα) in ER+ cells, and to BRD4-containing enhancers in triple-negative breast cancer (TNBC) cells. We also show a functional crosstalk between RING1B, FOXA1, and ERα in ER+ cells, resulting in an attenuated response to estrogen with RING1B depletion. We provide evidence that RING1B directly regulates chromatin accessibility at enhancers bound by transcription factors involved in breast cancer. In agreement with survival prognoses of patients with different breast cancer subtypes and *RNF2* expression levels, RING1B differentially regulates the metastatic potential of TNBC and ER+ breast cancer cells. Finally, we show that RING1B is recruited to enhancer regions in other cancer types, suggesting that this RING1B-mediated mechanism of controlling oncogenic pathways occurs in multiple cancers.

## Results

### cPRC1 genes are amplified and overexpressed in breast cancer.
To initially assess whether PRC1 components are altered in cancer, we examined the mutational frequencies of the histone H2A mono-ubiquitin ligases *RNF2* (encoding RING1B) and *RING1*, the cPRC1 genes, and the core PRC1-encoding genes (Supplementary Fig. 1a) in large-scale genomic data sets from cancer patients. We found that PRC1 genes were amplified in multiple cancer types. Intriguingly, many hormone-related cancers (e.g., ovarian, uterine, prostate, and breast cancer) were

overrepresented (Supplementary Fig. 1b). Since the breast cancer data sets contain the largest number of patient samples and thus provide the most robust data, we further analyzed PRC1 genes in these patient samples. We found that *RNF2* was amplified in up to 22% of breast cancers and cPRC1 genes were amplified in a large number of samples (Supplementary Fig. 1c–d). Compared to *RING1* which is not amplified, *RNF2* amplification correlated to its significant overexpression in breast cancer compared to normal breast tissues, regardless of breast cancer subtype (Supplementary Fig. 1e–f). We also noticed that other amplified cPRC1 genes, including *CBX2/4/8* and *PCGF2*, exhibited distinct expression patterns when categorized by breast cancer subtype (Supplementary Fig. 1g). Furthermore, *RNF2* expression was highest in tumors with amplification of the gene (Supplementary Fig. 2a). However, *RNF2*, *PCGF2*, and *CBX2/4/8* expression was higher in all four breast cancer stages compared to normal breast tissue, suggesting that their overexpression was not predictive of breast cancer aggressiveness (Supplementary Fig. 2b).

### RING1B binding is redistributed in breast cancer cells.
We next focused on understanding the specific role of RING1B in breast cancer (Fig. 1a). To our knowledge, no genome-wide study of RING1B binding to chromatin in breast cancer cells had yet been conducted. We performed RING1B chromatin immunoprecipitation followed by massive parallel sequencing (ChIP-seq) of two breast cancer cell lines—estrogen receptor positive (ER+) luminal A cell line, T47D, and triple-negative breast cancer (TNBC) cell line, MDA-MB-231—and a non-tumorigenic transformed mammary epithelial cell line, MCF10A. As a control, we also performed RING1B ChIP-seq in human induced pluripotent stem cells (iPSCs) since the target genes of PRC1 subunits have been extensively mapped in stem cells[16,17]. Additionally, the RING1B antibody used is validated by mass spectrometry. To further confirm the specificity of this antibody, we performed RING1B western blotting and immunoprecipitation from control and RING1B-depleted MDA-MB-231 cells (Supplementary Fig. 3a–b). As additional controls, we performed ChIP-qPCR of known RING1B target genes in iPSCs[17] using a different RING1B antibody as well as H3K27me3, H3K4me3 and H3K27ac antibodies (Supplementary Fig. 3c–d) and the enrichment values are in agreement with ChIP-seq binding.

We identified 702 RING1B target genes in iPSCs, 2869 in MCF10A, 2202 in T47D, and 2137 in MDA-MB-231 (Fig. 1b and Supplementary Data 1). Gene ontology (GO) analyses revealed RING1B targets as developmental genes in iPSCs (Fig. 1c), in agreement with published data[17]. In contrast, GO analysis of RING1B targets in MCF10A showed enrichment of genes involved in axon guidance and focal adhesion, while in T47D and MDA-MB-231, genes involved in focal adhesion, cell-to-cell junctions, and signaling pathways in cancer were enriched (Fig. 1c). As expected based on the GO analyses, the overlap of RING1B targets was relatively low between iPSCs, MCF10A, T47D, and MDA-MB-231 (Fig. 1d), but higher between MCF10A, T47D, and MDA-MB-231 (Supplementary Fig. 3e–f).

To determine the functional significance RING1B genomic distribution, we categorized RING1B ChIP-seq peaks into three main regions: intergenic, intragenic, and promoter regions (Methods section). Most RING1B peaks in iPSCs were located at promoters or inside genes. However, in MCF10A, T47D, and MDA-MB-231, RING1B was distributed to intergenic regions (Fig. 1e). We also found that each of the cell lines had a set of distinct RING1B peaks corresponding to cancer-related and epithelial genes in the breast cells but not in iPSCs (Fig. 1f, g and Supplementary Fig. 3g). Importantly, RNA-seq analysis indicated that RING1B target genes in MCF10A, T47D, and MDA-MB-231

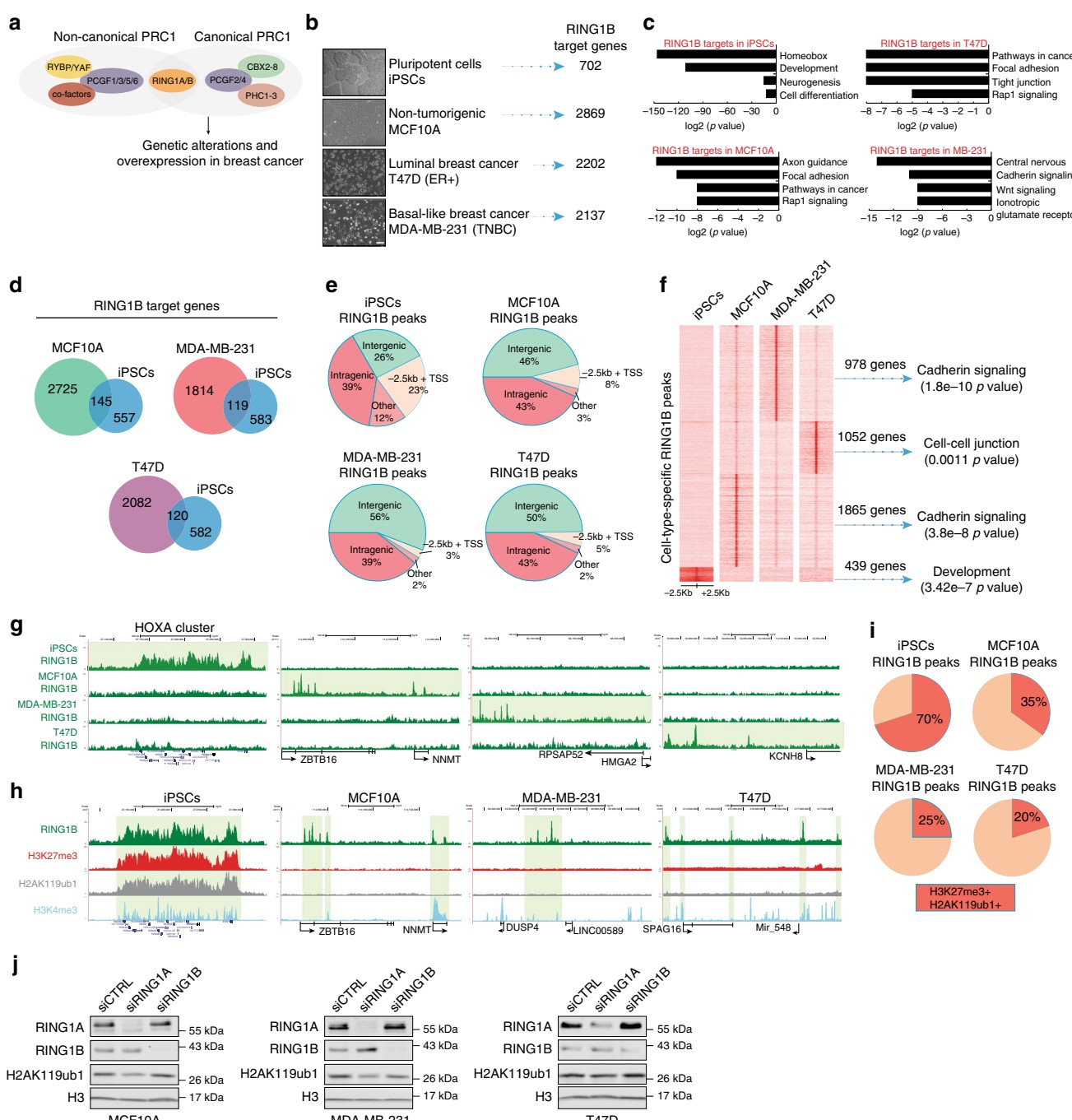

**Fig. 1** Genome-wide occupancy and activity of RING1B in breast cancer cells. **a** Model depicting RING1B and cPRC1 subunits that are genetically amplified and overexpressed in breast cancer. **b** Number of RING1B target genes. Representative phase-contrast images of each cell line are shown at ×10 magnification. Scale bar represents 100 μm. **c** GO analysis of RING1B target genes. **d** Venn diagrams of overlapping RING1B target genes. **e** Distribution of RING1B ChIP-seq peaks. **f** ChIP-seq heat maps of specific RING1B peaks in each of the cell lines. GO analysis performed on target genes identified in each peak cluster. **g** Genome browser screenshots of unique RING1B-binding sites in each of the cell lines. RING1B peaks are highlighted in green. **h** Pie chart showing percentage of RING1B peaks overlapping with H2AK119ub1 and H3K27me3. **i** Genome browser screenshots of RING1B, H3K27me3, H2AK119ub1, and H3K4me3 in each of the cell lines. RING1B peaks are highlighted in green. **j** Representative western blots of RING1A, RING1B, and H2AK119ub1 of control and RING1B-depleted cells. Histone H3 was used as a loading control ($n = 3$)

are transcriptionally more active and more highly expressed than the RING1B target genes in iPSCs (Supplementary Fig. 3h–i).

**Most RING1B-bound sites are devoid of H3K27me3/H2AK119ub1.** Since the classical model of PRC1 binding to chromatin is following PRC2 recruitment, we next determined

the degree of overlap between RING1B and the PRC2-associated and PRC1-associated histone modifications, H3K27me3 and H2AK119ub1, respectively. As expected in iPSCs, the majority of sites containing RING1B were also decorated with both histone modifications (Fig. 1h, i and Supplementary Fig. 3j)[18]. In MCF10A, 35% of RING1B sites were co-occupied by H3K27me3 and H2AK119ub1 and this overlap decreased to 25 and 20% in

MDA-MB-231 and T47D, respectively (Fig. 1h, i and Supplementary Fig. 3k–l). This observation was confirmed by overlapping RING1B target genes and H3K27me3-marked genes (Supplementary Fig. 3m). These results indicate that in breast epithelial cells: (1) RING1B function is not exclusively associated to its mono-ubiquitination ligase activity and (2) RING1B is recruited to chromatin independently of PRC2. In agreement with the low overlap between RING1B and H2AK119ub1, RING1B depletion had no major effect on bulk levels of H2AK119ub1. However, H2AK119ub1 levels were reduced after RING1A depletion in MDA-MB-231 and T47D (Fig. 1j), indicating that RING1A enzymatic activity at histone H2A is greater than RING1B in these cells. This line of evidence suggests that RING1A is the main histone H2A mono-ubiquitin ligase in these breast cancer cell lines.

**RING1B binds active enhancers**. Since a large number of RING1B sites were not marked with H3K27me3 or H2AK119ub1 and RING1B peaks re-localized to intergenic regions, we next tested whether RING1B is recruited to enhancer regions. Enhancers are regulatory sites that can be bound by transcription factors to increase the transcription of a particular gene[19,20] and can be divided into typical enhancers and super-enhancers (SEs): in cancer, typical enhancers promote transcription at active genes and SEs regulate the expression of oncogenes and genes associated to oncogenic transcriptional programs[21]. Active typical enhancers and SEs are also epigenetically distinct: although both are marked with H3K4me1, SEs contain increased levels of H3K27ac[21,22]. We found that both H3K27ac and H3K4me1 ChIP-seq signals were enriched at RING1B-bound sites (Fig. 2a) that were simultaneously devoid of H3K27me3 (Supplementary Fig. 4a–b). Only 4%, 8%, and 13% of typical enhancers contained RING1B in MCF10A, MDA-MB-231, and T47D cells, respectively, while in contrast, over 45% of SEs in these cells were decorated with RING1B (Fig. 2b–d and Supplementary Fig. 4c–d). Virtually none of the SEs in iPSCs contained RING1B (Fig. 2c).

We next asked whether RING1B was recruited to SEs near genes with established functions in breast cancer. Indeed, we observed RING1B recruitment at SE regions near *BCL2L1* in MDA-MB-231 and *ESR1* in T47D[23,24] (Fig. 2e and Supplementary Data 1). To confirm that the SEs were unique to each cell line, and that RING1B was recruited specifically to these unique sites, we determined the RING1B signal at these SEs. We found that RING1B signal at MDA-MB-231 specific SEs was stronger in MDA-MB-231 cells than at the same SE regions in MCF10A and T47D cells; the same was true for MCF10A- and T47D-specific SEs (Fig. 2f and Supplementary Fig. 4e). These results indicate that RING1B is recruited to cell-type-specific SEs in breast epithelial and cancer cells.

In contrast to the broad RING1B ChIP-seq signals in pluripotent cells, RING1B peaks in the breast cell lines were narrow (Figs. 1g, h and 2e), resembling ChIP-seq signals of transcription factors. Therefore, we assessed whether RING1B is recruited to specific transcription factor-binding sites at SEs[20]. In the ER + cell line, T47D, analysis of known transcription factor motifs revealed an enrichment of the ERα and FOXA1/2 consensus binding sequences[25,26] (Fig. 2g), suggesting a functional connection between RING1B and the ER pathway. Similarly, motifs for important breast cancer oncogenic transcription factors were overrepresented at RING1B-containing SEs in MDA-MB-231 and MCF10A cells that are ER− (Fig. 2g).

Finally, we associated potential target genes to the SEs containing RING1B based on proximity and retrieved 561, 252, and 398 genes that were potentially functionally associated with

SEs in MCF10A, MDA-MB-231, and T47D cells, respectively (Fig. 2h and Supplementary Data 2). Interrogation of published ChIP-seq data sets in ENCODE using EnrichR revealed a further potential functional association of RING1B with ERα in T47D. Interestingly, the bromodomain-containing protein, BRD4, was recruited to genes potentially controlled by RING1B-containing SEs in MDA-MB-231, while RACK7 (receptor for activated C-kinase 7) bound the RING1B-containing SEs in MCF10A (Fig. 2h). Overall, these results indicate that RING1B is recruited to SEs and, importantly, that there is a specific functional crosstalk between RING1B and key signaling pathways involved in breast cancer.

**RING1B assembles into discrete cPRC1 complexes**. Dozens of cPRC1 and ncPRC1 variants can be potentially assembled, and have distinct biological functions in regulating stem cell pluripotency, differentiation, and tissue homeostasis[3,6,27–30]. To assay the RING1B protein interactome in MDA-MB-231 and T47D, we performed co-immunoprecipitations of endogenous RING1B-associated protein complexes using the anti-RING1B antibody used for ChIP-seq, followed by label-free quantitative liquid chromatography-tandem mass spectrometry (LC-MS/MS). Unexpectedly, because both cPRC1 and ncPRC1 genes are expressed in these cells (Supplementary Fig. 5a), RING1B mainly co-immunoprecipitated with cPRC1 subunits (Fig. 3a, b and Supplementary Data 3). Specifically, when captured from T47D cells, RING1B demonstrated interactions with the cPRC1 subunits CBX4/8, PCGF2, and PHC2/3 (Fig. 3a, left), with CBX8 and PHC3 displaying the highest levels of interaction with RING1B (Fig. 3b, left). RING1B co-immunoprecipitated a larger number of proteins in MDA-MB-231 cells than in T47D cells (Fig. 3a, right), but of the proteins observed, cPRC1 subunits, including CBX8, PCGF2, and PHC2 were amongst the most abundant (Fig. 3b, right). We next addressed whether the RING1B recruited to chromatin in T47D and MDA-MB-231 is a part of a cPRC1 complex. We performed ChIP-seq of PCGF2 since it is the predominant RING1B-associated PCGF subunit in both cell lines and identified 2408 and 4813 PCGF2 target genes in T47D and MDA-MB-231 cells, respectively (Fig. 3d, g). Almost 60% of PCGF2 targets in T47D cells, and about 80% in MDA-MB-231 cells, were also co-occupied by RING1B (Supplementary Fig. 5b–c).

To further interrogate the potential functional relationship between RING1B and ERα, we addressed whether a cPRC1 complex (defined by co-occupancy of RING1B and PCGF2) is associated with genomic sites bound by ERα. We found that cPRC1 was indeed co-recruited with ERα to a large number of genomic sites (Fig. 3c). Overlapping RING1B, PCGF2, and ERα targets indicated that 890 target genes were decorated with cPRC1 and ERα (Fig. 3d). Importantly, these genes are involved in pathways important in carcinogenesis (Fig. 3e and Supplementary Data 4). Furthermore, since we observed that potential genes regulated by RING1B-containing SEs in MDA-MB-231 cells were BRD4 targets (Fig. 2h), we also performed ChIP-seq of BRD4. Indeed, cPRC1 largely associated with BRD4 targets and 840 genes were decorated with cPRC1/BRD4 (Fig. 3f, g and Supplementary Data 4). These cPRC1/BRD4 co-targets are involved in cancer and focal adhesion pathways (Fig. 3h).

We next determined the co-recruitment of cPRC1 with either ERα or BRD4 to enhancers in T47D and MDA-MB-231 cells, respectively (Fig. 3i–k and Supplementary Fig. 5d). A total of 81% and more than 90% of SEs with cPRC1 were also bound by ERα in T47D and BRD4 in MDA-MB-231, respectively (Fig. 3l). Association of BRD4 to RING1B-containing enhancers in MDA-MB-231 was further validated by ChIP-qPCR (Supplementary

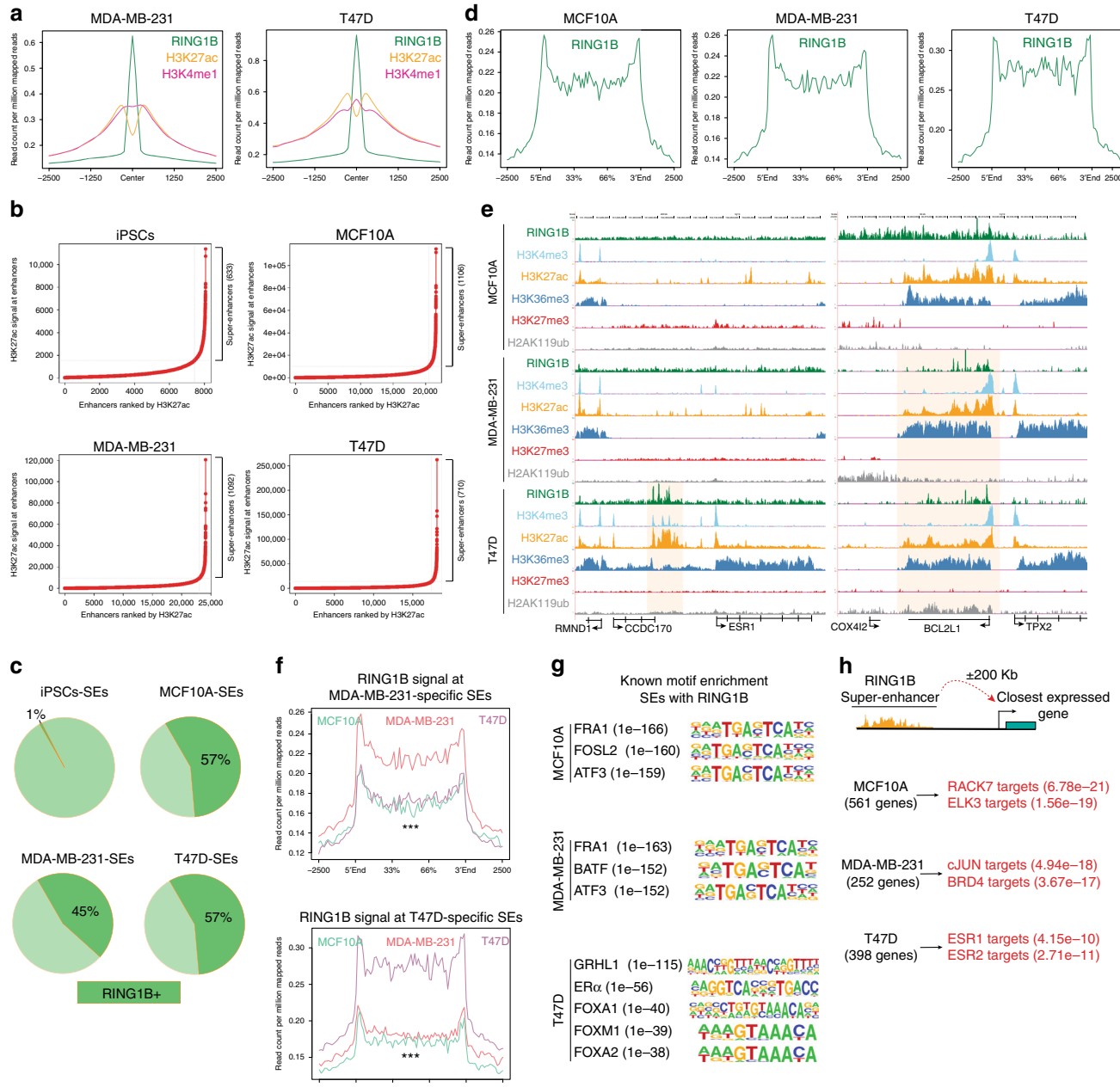

**Fig. 2** RING1B is recruited to super-enhancers. **a** H3K27ac and H3K4me1 ChIP-seq signals relative to RING1B peak summit. **b** Super-enhancers (SEs) identified in each cell line. **c** Pie charts showing percentage of SEs containing RING1B. **d** RING1B ChIP-seq signal at SEs. **e** Genome browser screenshots of RING1B and histone modifications. SE regions near *ESR1* and *BCL2L1* are highlighted in yellow. **f** RING1B ChIP-seq signal at T47D-specific SEs (top) or MDA-MB-231–specific SEs (bottom). RING1B ChIP-seq signal in RING1B-T47D SEs compared to RING1B ChIP-seq signal in the same genomic region in MDA-MB-231 (p-value = 3.07e − 24) and MCF10A (p-value = 2.39e − 29). RING1B ChIP-seq signal in RING1B-MDA-MB-231 SEs compared to RING1B ChIP-seq signal in the same genomic region in T47D (p-value = 1.15e − 16) and MCF10A (p-value = 1.36e − 09). Significance was determined by the Kolmogorov–Smirnov test. ***p-value < 0.001. **g** Transcription factor motif analysis of SEs containing RING1B. **h** Enrichr analysis of ENCODE ChIP-seq data using the nearest genes from SEs containing RING1B

Fig. 5e). We conclude that cPRC1 complexes are co-recruited to genes and enhancers targeted by key factors that regulate transcriptional networks in breast cancer.

**RING1B regulates oncogenic pathways and enhancer RNAs.**
Next, we determined the effects of RING1B depletion on gene expression in T47D and MDA-MB-231 cells. We found that in T47D, more genes were downregulated than upregulated (62% versus 38%) after RING1B depletion, suggesting that RING1B

facilitates gene activation (Fig. 4a, left, and Supplementary Fig. 6a). In contrast, in MDA-MB-231, RING1B depletion had a more modest effect on gene regulation as only about 90 genes were significantly deregulated (Fig. 4a, right). Deregulated genes in both cell lines included key genes involved in breast cancer progression and metastasis (Fig. 4b and Supplementary Fig. 6b). Additionally, these deregulated genes were significantly enriched as ERα targets in T47D cells (Supplementary Fig. 6c). *CD36*, which regulates fatty acid metabolism and metastasis[31], was the second most upregulated gene in

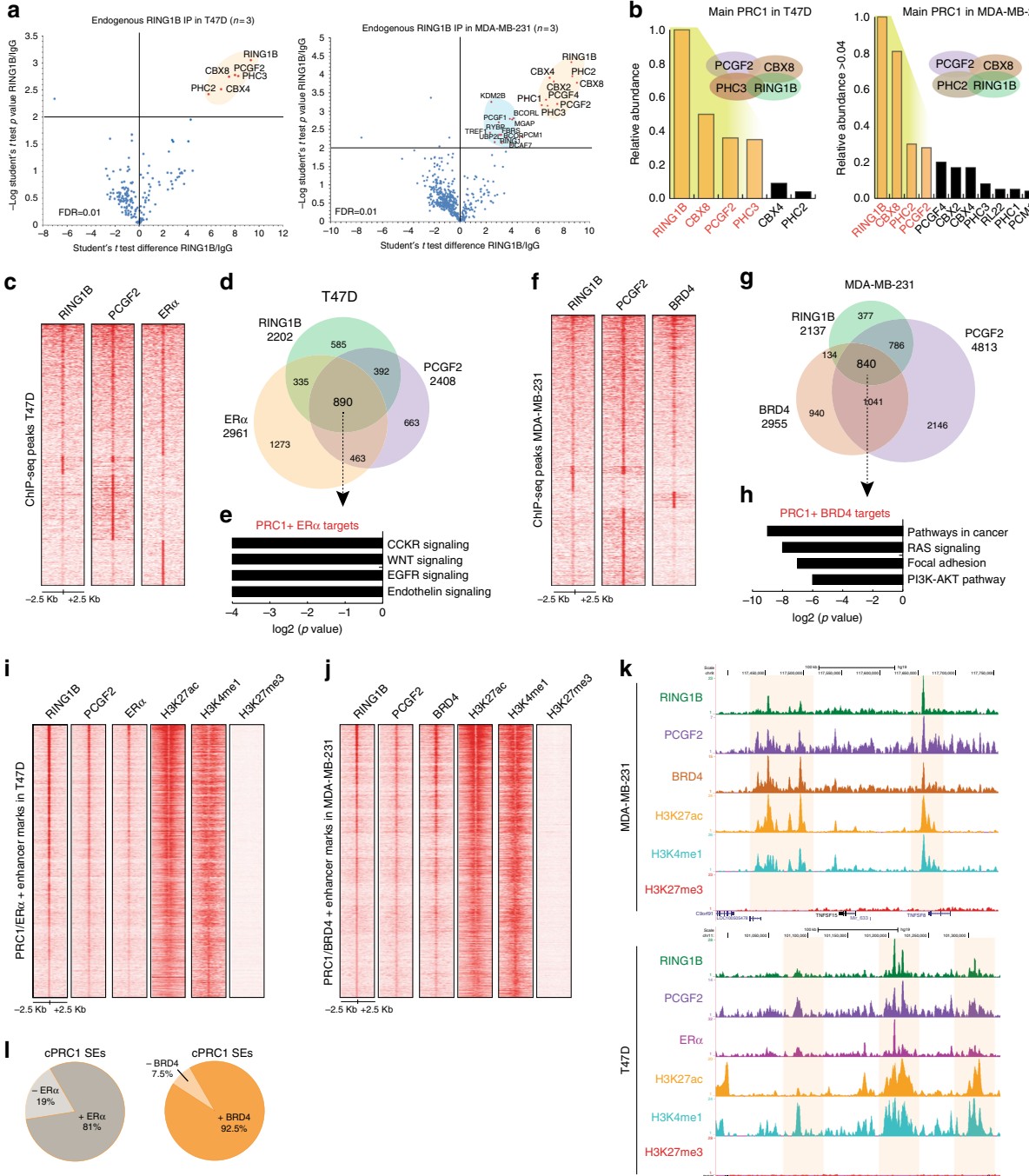

**Fig. 3** The RING1B interactome and its genome-wide association with ERα and BRD4 in ER+ and TNBC cells. **a** Endogenous RING1B immunoprecipitation with whole-cell extracts. Proteins bound to RING1B were identified by LC-MS/MS, and enrichment was calculated based on LFQ intensities. IgG was used as a negative control. Experiments were performed in three biological replicates. **b** Relative abundance of RING1B interactors. **c** ChIP-seq heat maps of RING1B, PCGF2, and ERα in T47D. **d** Overlapping of RING1B, PCGF2, and ERα target genes in T47D. **e** GO analysis of RING1B/PCGF2/ERα co-target genes. **f** ChIP-seq heat maps of RING1B, PCGF2, and BRD4 in MDA-MB-231. **g** Overlapping of RING1B, PCGF2, and BRD4 target genes in MDA-MB-231. **h** GO analysis of RING1B/PCGF2/BRD4 co-target genes. **i–j** ChIP-seq heat maps of RING1B, PCGF2, ERα, and histone modifications associated with active enhancers and SEs in T47D, and PCGF2, BRD4 in MDA-MB-231. **k** Genome browser screenshots of SEs. SE regions are highlighted in yellow. **l** Pie charts of cPRC1-SEs with ERα in T47D and BRD4 in MDA-MB-231

shRING1B T47D (Fig. 4a, left and Supplementary Fig. 6b) and the fatty acid metabolism pathway was upregulated after RING1B depletion (Fig. 4b, left, Supplementary Fig. 6b, and Supplementary Data 5). In RING1B-depleted MDA-MB-231 cells, several well-known oncogenic signaling pathways were also deregulated after RING1B depletion (Fig. 4b, right and Supplementary Fig. 6b). RT-qPCR of select cancer-related genes

in both shRING1B T47D and MDA-MB-231 cells confirmed the RNA-seq results, and further suggested that fatty acid metabolism (represented by *CD36* and *HMGCS2*) may play a major role in the tumorigenesis of ER+ breast cancer (Fig. 4c). Although *RNF2* amplification did not correlate with overexpression in patients with HER2+ tumors, *RNF2* expression was significantly elevated compared to normal breast tissues

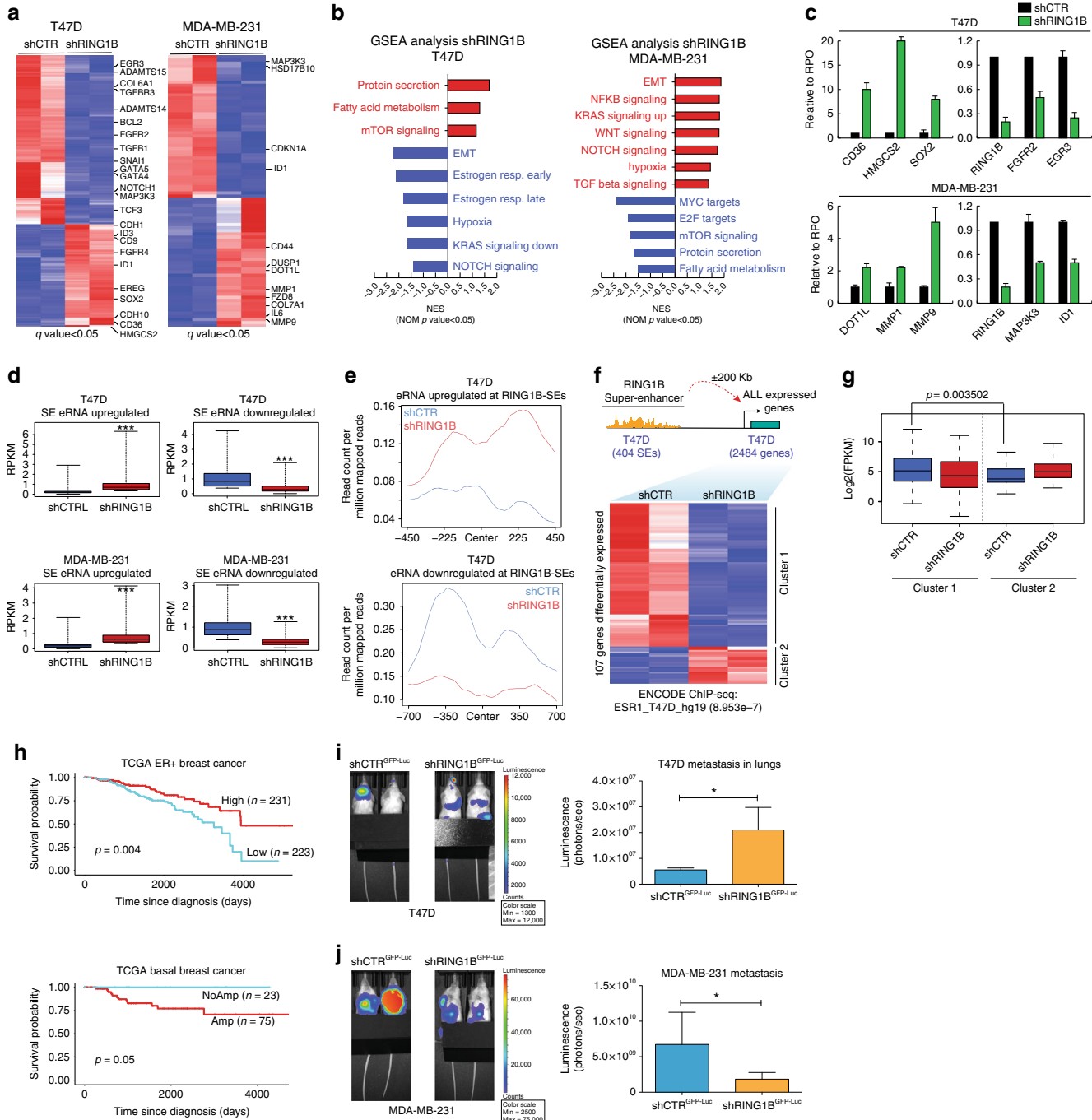

**Fig. 4** RING1B regulates specific oncogenic pathways and metastasis in breast cancer subtypes. **a** RNA-seq heat maps of upregulated and downregulated genes in RING1B-depleted (shRING1B) T47D and MDA-MB-231 cells. RNA-seq experiments were performed in two biological replicates. **b** GSEA analyses after RING1B depletion. **c** Real-time qPCR of selected genes in control and RING1B-depleted T47D and MDA-MB-231. Expression was normalized to the housekeeping gene RPO. Data represent the average of two independent experiments. **d** Box plots of deregulated eRNAs in SEs after RING1B depletion. **e** eRNA signal at RING1B-occupied SE regions. **f** Heatmap of deregulated genes near SEs containing RING1B in shRING1B T47D. **g** Genes in **f** that are downregulated (cluster 1) or upregulated (cluster 2) in shRING1B T47D. **h** Kaplan–Meier survival analysis of patients from TCGA with ER+ tumors (top) or Basal (TNBC) breast cancer tumors segregated by *RNF2* expression. **i–j** Representative images of metastatic signal detected by IVIS in NSG mice 65 days after injection of control and shRING1B T47DGFP-luc and MDA-MB-231GFP-luc cells in the mammary fat pad (*n* = 5/group). Quantification of luciferase signal by IVIS in control and shRING1B T47DGFP-luc and MDA-MB-231GFP-luc cells. Error bars represent SD. *p-value < 0.05; ***p-value < 0.001, two-tailed *t*-test. Center line of box plots represent the median and upper and lower bounds of whiskers represent the maximum and minimum values, respectively

(Supplementary Fig. 2a). To assess whether RING1B depletion also affected oncogenic pathways in HER2+ cells, we stably depleted RING1B in the commonly used HER2+ cell line, SKBR3, and performed RNA-seq experiments (Supplementary Fig. 6d). A total of 674 genes were deregulated (*q*-value < 0.05,

fold change > 2) upon RING1B KD, with 255 and 419 genes upregulated and downregulated, respectively (Supplementary Fig. 6e), suggesting that RING1B may also positively regulate gene expression in HER2+ cells. GSEA analysis revealed a strong deregulation of cancer-related pathways, including cell

cycle, TGF-β and PPAR signaling, and fatty acid metabolism (Supplementary Fig. 6e–f).

Since RING1B was bound to enhancers, we next asked whether RING1B depletion affected the expression of enhancer RNAs (eRNAs)[32]. RING1B depletion significantly dysregulated eRNAs transcribed from active typical enhancers and SEs (Fig. 4d and Supplementary Fig. 6g). Importantly, RING1B was recruited to 64 and 53% of SE eRNAs that were differentially expressed after RING1B depletion in both cell lines (Fig. 4e, Supplementary Fig. 6h).

Finally, we assessed whether RING1B depletion affected expression of genes potentially regulated by RING1B-containing SEs (as identified in Fig. 2h). In T47D, of the 2484 genes identified that are potentially regulated by 404 RING1B-SEs, 107 were deregulated upon RING1B depletion. Although most were downregulated (cluster 1) and included important genes for breast epithelial homeostasis (e.g., *LRIG1*, *CYP27B1*, *HES1*, *THBS1*), a set of genes were upregulated (cluster 2) (Fig. 4f). These results suggest that at enhancer regions, RING1B potentially plays a dual function in gene expression (cluster 1) and gene repression (cluster 2) (Fig. 4g).

**Role of RING1B in breast cancer tumorigenesis and metastasis**. We next sought to determine the function of RING1B in breast cancer tumorigenesis and metastasis in vivo. We hypothesized that RING1B depletion increases the aggressiveness of T47D cells due to the strong upregulation of *CD36*, a marker for metastasis-initiating cells (Fig. 4a, c and Supplementary Fig. 7a). However, RING1B depletion in MDA-MB-231 cells resulted in both positive and negative deregulation of genes involved in breast cancer, thus we could not anticipate the role of RING1B in TNBC in vivo. Our initial analysis of the TCGA breast cancer data set (Fig. 4h) indicated that patients with ER+ breast cancer and high levels of *RNF2* survive longer than patients with lower *RNF2* levels. In contrast, patients with basal breast cancer and high levels of *RNF2* have a lower survival probability. This data suggested that RING1B might exert divergent functions in tumor formation or metastasis in specific breast cancer subtypes. To assess whether T47D and MDA-MB-231 cells recapitulate the results obtained with the TCGA data set, we injected control and shRING1B cells into the mammary fat pad of NSG mice . Cells were engineered to express a GFP-luciferase transgene to monitor tumor formation and metastasis by IVIS (Supplementary Fig. 7b). Although we did not detect significant changes in primary tumor development between control and RING1B-depleted T47D and MDA-MB-231 cells (Supplementary Fig. 7c–d), mice with tumors derived from T47D-shRING1B cells lost more weight than control animals. In contrast, mice with tumors derived from shRING1B-MDA-MB-231 were heavier than control animals (Supplementary Fig. 7e). T47D cells are not highly metastatic[33], yet shRING1B T47D but not control cells metastasized to the lungs (Fig. 4i). In the highly metastatic MDA-MB-231 tumors[33], depletion of RING1B reduced the metastatic potential of these cells (Fig. 4j). Importantly, these results are in agreement with our TCGA survival analysis (Fig. 4h), and further support the concept of RING1B being a pro-metastatic gene in basal breast cancer and a suppressor of metastasis in ER+ tumors.

**A novel RING1B-FOXA1-ERα transcriptional axis in ER+ cells**. In T47D cells, RING1B was recruited to SEs containing FOXA1 and ERα-binding sites (Figs. 2g and 3c, i, l). Among those, RING1B bound to the SE that regulates *ESR1* (encoding ERα) (Fig. 2e). Moreover, RING1B depletion strongly affected the "Estrogen Response" gene signature (Fig. 4b). These results suggested that RING1B is functionally involved in the estrogen

signaling pathway through an ERα/FOXA1 transcriptional regulatory axis. Interestingly, in MDA-MB-231 cells that do not express *FOXA1*, RING1B was recruited to the *FOXA1* promoter and had a canonical repressive function, co-localizing with H2AK119ub1 and H3K27me3 histone marks (Fig. 5a). In contrast, in T47D cells, RING1B bound to a putative SE downstream of *FOXA1*, suggesting that it plays an activating role in regulating *FOXA1* expression (Fig. 5a). RING1B ChIP-qPCR of several RING1B-SEs in control and RING1B-depleted T47D cells confirmed the binding of RING1B to enhancer regions identified by ChIP-seq, including the *FOXA1* putative enhancer (Fig. 5b and data not shown). We then assessed whether RING1B directly regulates *FOXA1* expression in both T47D and MDA-MB-231 cells. While RING1B depletion in MDA-MB-231 was not sufficient to activate *FOXA1* expression (data not shown), acute depletion of RING1B by siRNA reduced FOXA1 protein levels ~50% in T47D cells (Fig. 5c, left panel). Although FOXA1 levels remained unaffected upon stable RING1B depletion by shRNA (Fig. 5c, right panel), cellular fractionation assays showed that FOXA1 was displaced from chromatin and relocated to the soluble nuclei fraction (Fig. 5d). Since FOXA1 is a transcription factor important for ERα recruitment to chromatin[26], displacement of FOXA1 from chromatin also impaired chromatin localization of ERα (Fig. 5d). This set of data suggests that RING1B mediates the estrogen response by affecting FOXA1 and ERα recruitment to chromatin.

We then asked whether FOXA1 depletion affected RING1B levels. While acute FOXA1 depletion affected the RING1B protein levels moderately (Fig. 5e, left panel), stable FOXA1 depletion strongly reduced RING1B global levels (Fig. 5e, right panel). Importantly, RING1B binding to chromatin was also severely reduced (Fig. 5f). Analysis of FOXA1 ChIP-seq in T47D cells did not reveal binding of FOXA1 to the *RNF2* promoter (data not shown).

Finally, since we observed reduced levels of both FOXA1 and ERα at chromatin upon RING1B depletion, we asked whether RING1B-depleted cells can respond to estrogen stimulation. To this end, we cultured control and RING1B KD cells for 72 h in hormone-deprived (HD) media prior to induction of ERα signaling with 10 nM of E2 (estradiol) for 12h[34]. In agreement with the global gene expression profiles of RING1B-depleted T47D cells (Fig. 4b), there was reduced expression of prominent E2-responsive genes in shRING1B T47D compared to control cells (Fig. 5g). Altogether, these results demonstrated that RING1B is a novel epigenetic factor that directly and indirectly regulates the FOXA1–ERα axis by multiple mechanisms (Fig. 5h).

**RING1B regulates chromatin accessibility at enhancers**. Since RING1B was recruited to regions targeted by transcription factors and its depletion deregulated breast cancer signaling pathways as well as FOXA1 and ERα localization to chromatin, we next hypothesized that RING1B regulates transcriptional programs in breast cancer by orchestrating chromatin accessibility. To test this, we performed transposase-accessible chromatin sequencing (ATAC-seq)[35] in RING1B-depleted cells (Fig. 6a). As expected, ATAC-seq peaks in control cells were at promoter, intronic, and intergenic regions (Supplementary Fig. 8a). Importantly, ATAC-seq peaks co-localized with a large number of RING1B peaks in control cells, and the majority of this co-localization occurred at introns and intergenic regions, but not at promoters (Fig. 6b). These results indicate that RING1B depletion affects chromatin accessibility at enhancer regions.

We next asked whether RING1B depletion-induced de novo generation and/or loss of accessibility sites. RING1B depletion generally affected chromatin accessibility, suggesting that

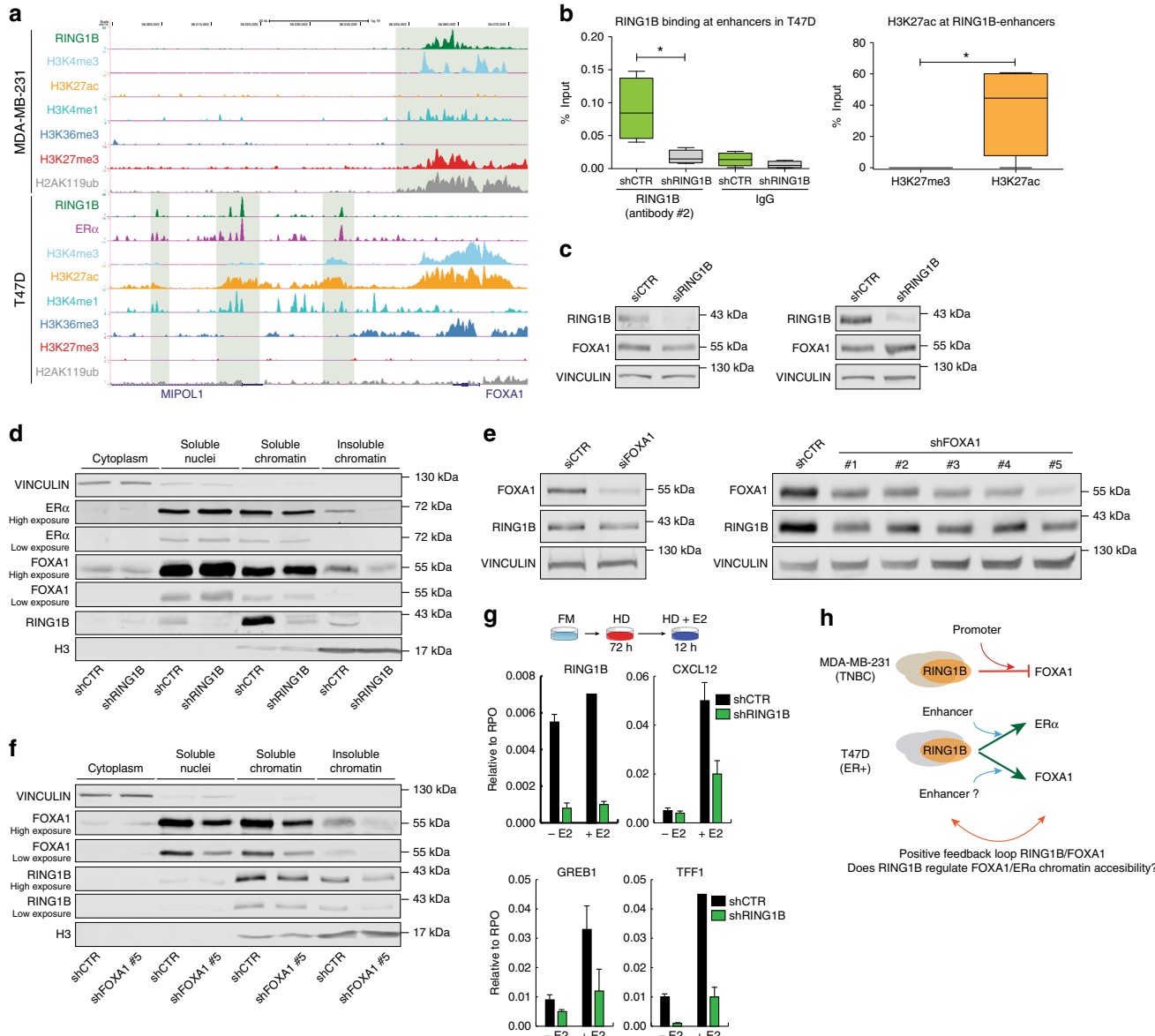

**Fig. 5** RING1B regulates FOXA1 and ERα through multiple mechanisms. **a** Genome browser screenshots of the profiles of RING1B and histone modifications in T47D and MDA-MB-231 cells at the *FOXA1* locus. **b** RING1B, H3K27me3 and H3K27ac ChIP-qPCR of RIN1GB-containing enhancers in control and RING1B-depleted T47D cells. IgG antibody was used as a negative control. As additional control, RING1B ChIP-qPCR were performed using a different RING1B antibody from the one used for ChIP-seq. Error bars represent the SD of two independent experiments. *p-value < 0.05, two-tailed t-test. **c** Western blot of RING1B and FOXA1 from control and RING1B-depleted cells 72 h after siRNA transfection (left panel) or after puromycin selection of shRING1B T47D cells (right panel). VINCULIN was used a loading control. **d** Western blot of RING1B, FOXA1 and ERα after cellular fractionation of control and RING1B-depleted T47D cells. VINCULIN and histone H3 were used as a cytoplasmic and chromatin fraction controls, respectively. **e** Western blot of RING1B and FOXA1 from control and FOXA1-depleted cells 72 h after siRNA transfection (left panel) or after puromycin selection of shFOXA1 T47D cells (right panel). VINCULIN was used a loading control. **f** Western blot of RING1B and FOXA1 after cellular fractionation of control and FOXA1-depleted T47D cells. VINCULIN and histone H3 were used a cytoplasmic fraction and chromatin fraction control, respectively. All the cellular fractionation experiments and total protein extracts shown in the figure were performed at least three times. **g** RT-qPCR of E2-responsive genes in control and RING1B-depleted T47D after administration of E2 (10 mM) for 12 h in cells cultured in hormone-deprived (HD) media for 72 h. FM full media. Error bars represent SD of two independent experiments. **h** Model of RING1B action in MDA-MB-231 and T47D cells

RING1B is involved in both opening and closing chromatin (Fig. 6c, d). Upon RING1B depletion, the ATAC-seq peaks either lost or gained de novo were located at introns and intergenic regions (Supplementary Fig. 8b–c). Notably, RING1B was recruited to genomic regions not accessible to transposase in control cells but became accessible in RING1B-depleted cells (Fig. 6e, f, top). Further, RING1B was recruited to open chromatin sites and its depletion-induced chromatin compaction

(Fig. 6e, f, bottom). These results suggest that RING1B plays a dual role in regulating chromatin accessibility.

We next analyzed the impact of RING1B depletion on chromatin accessibility at enhancers. In T47D cells, RING1B depletion resulted in the loss of about 500 peaks and gain of more than 600 de novo peaks at enhancers (Fig. 6g). RING1B binds to 55% of SEs and 23% of typical enhancers (Fig. 6g). Transcription factor motif analysis revealed that ATAC-seq peaks lost at

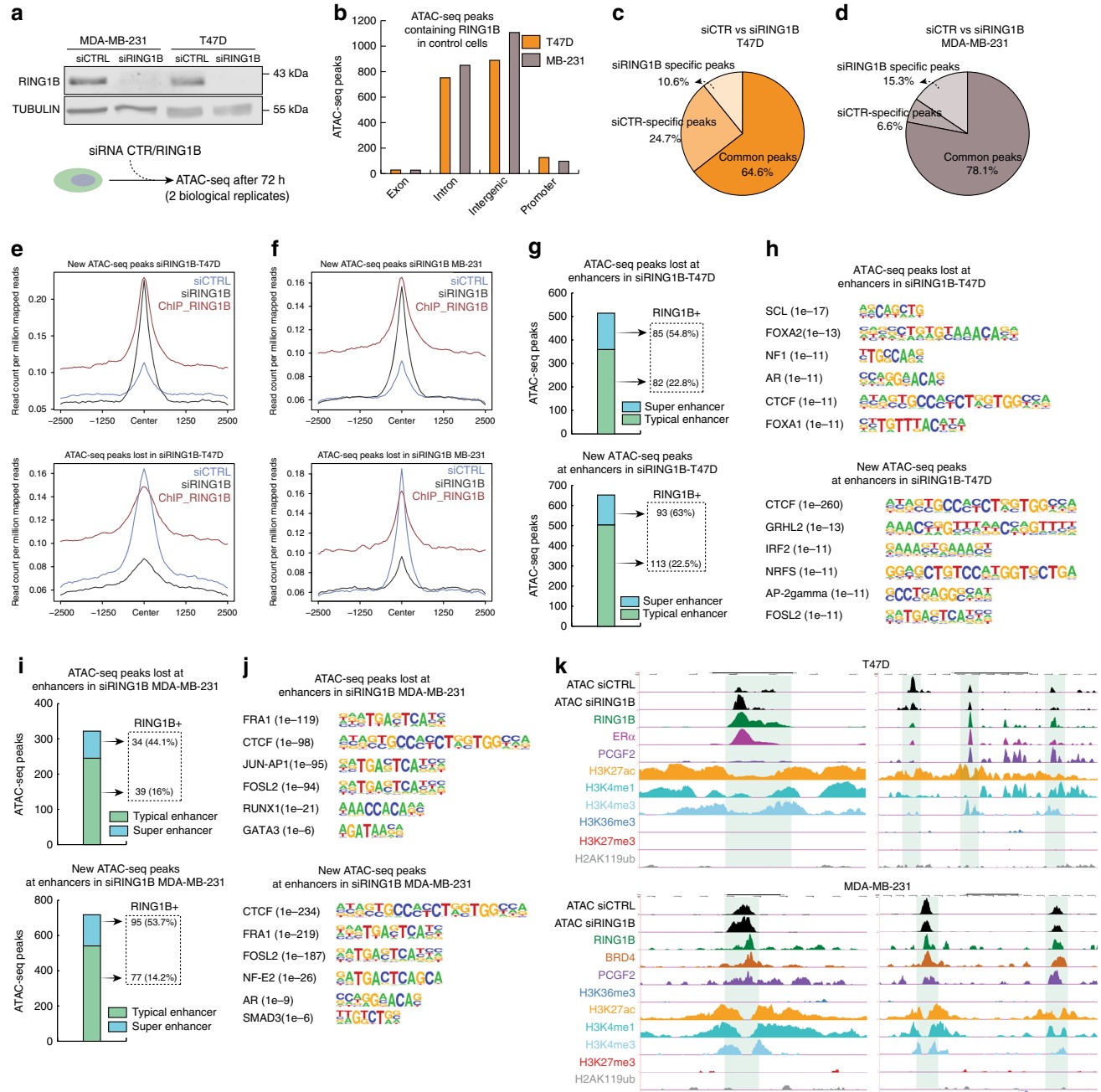

**Fig. 6** RING1B regulates chromatin accessibility at enhancers. **a** Western blot of RING1B from control and RING1B-depleted cells 72 h after siRNA transfection. TUBULIN was used a loading control. ATAC-seq experiments were performed in two biological replicates after siRNAs transfections. **b** ATAC-seq peak distribution in genomic sites bound by RING1B in RING1B-depleted cells. **c, d** Pie charts showing percentage of ATAC-seq peaks not affected by RING1B depletion (common peaks), lost after RING1B depletion (siCTR-specific peaks), or gained after RING1B depletion (siRING1B-specific peaks. Two ATAC-seq experiments were performed after two independent siRING1B transfections. **e, f** RING1B ChIP-seq signals and ATAC-seq signals at acquired and lost ATAC-seq peaks. **g** ATAC-seq peaks at enhancers after RING1B depletion, number of enhancers containing RING1B ChIP-seq signals in T47D. **h** Transcription factor-binding motif analysis of peaks acquired or lost at enhancers in T47D. **i** Acquired and lost ATAC-seq peaks at enhancers after RING1B depletion, number of enhancers containing RING1B ChIP-seq signals in MDA-MB-231. **j** Transcription factor-binding motif analysis in peaks acquired or lost at enhancers in MDA-MB-231. **k** Genome browser screenshots of ChIP-seq and ATAC-seq profiles at selected enhancers

enhancer regions contained FOXA1/2-binding sites (Fig. 6h, top), further confirming a functional association between RING1B and ERα. In contrast, de novo ATAC-seq peaks in T47D-contained CTCF-binding sites, suggesting that RING1B might be involved in maintaining topological-associated domains (TADs)[36] (Fig. 6h, bottom).

The influence of RING1B on chromatin accessibility in MDA-MB-231 was less profound than in T47D (Fig. 6d), which is in line with the modest gene expression changes in shRING1B MDA-MB-231 cells. However, about 300 and 700 ATAC-seq peaks were lost and gained at enhancers, respectively, after RING1B depletion (Fig. 6i). Interestingly, in addition to CTCF sites, accessibility was altered for breast cancer-specific transcription factors (Fig. 6j). Furthermore, altered chromatin accessibility at enhancers were co-bound by cPRC1/ERα and cPRC1/BRD4 in T47D and MDA-MB-231 cells, respectively (Fig. 6k). Overall,

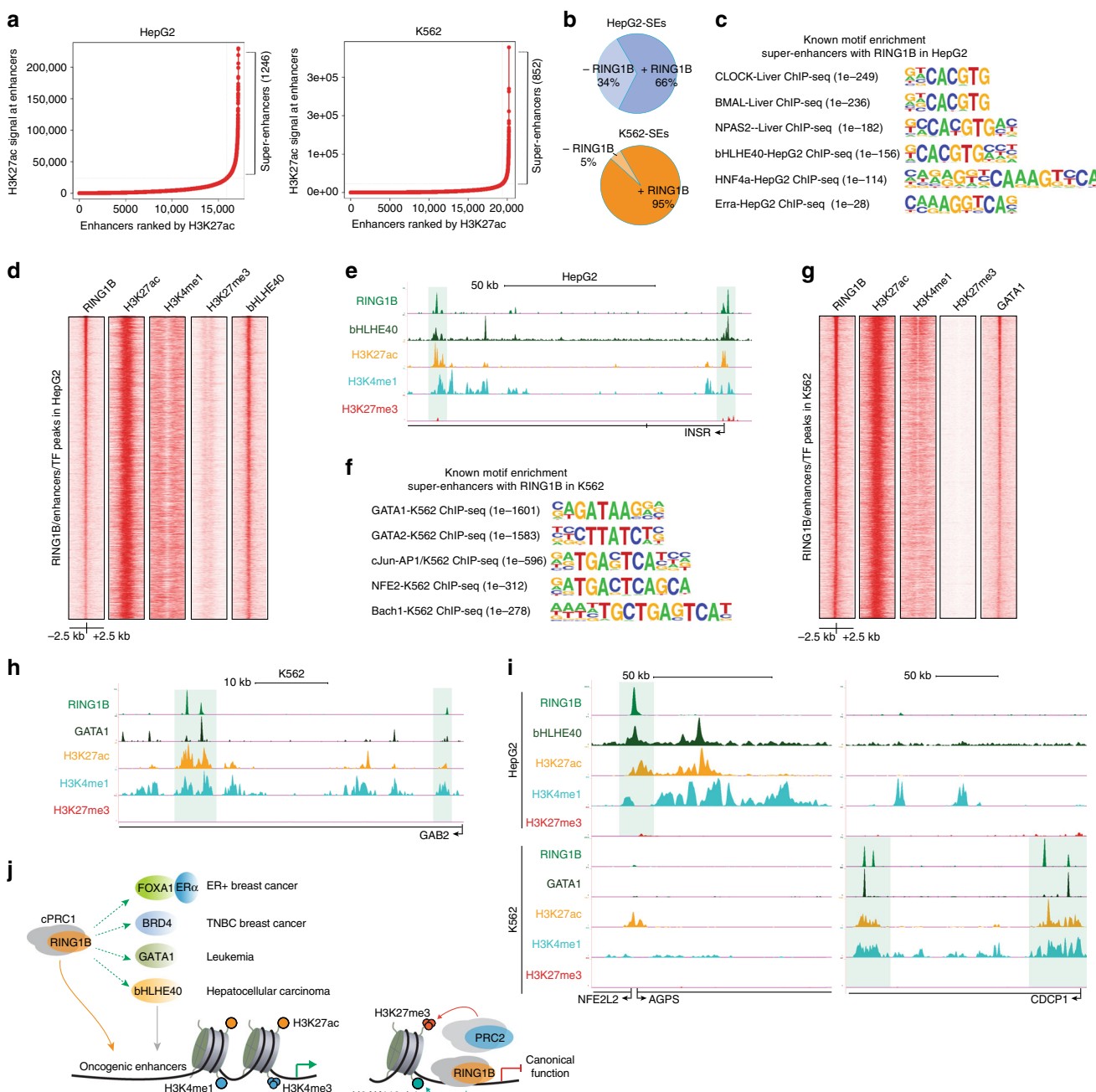

**Fig. 7** RING1B is recruited to super-enhancers in other cancer types. **a** SEs identified in HepG2 and K562 cells. **b** Percentage of SEs containing RING1B in HepG2 and K562. **c** Transcription factor-binding motif enrichment of SEs containing RING1B in HepG2 cells. **d** ChIP-seq heat maps of RING1B and histone modifications associated with active enhancers and SEs. **e** Genome browser screenshots of co-occupancy of RING1B and bHLHE40 at enhancers in HepG2. **f** Transcription factor-binding motif enrichment of SEs containing RING1B in K562 cells. **g** ChIP-seq heat maps of RING1B and histone modifications associated with active typical enhancers and SEs, and GATA1 in K562 cell lines. **h** Genome browser screenshots of co-occupancy of RING1B and GATA1 at enhancers in K562. **i** Genome browser screenshots of RING1B and GATA1 or RING1B and bHLHE40 co-occupancy at specific SEs in K562 and HepG2 cells, respectively. **j** Model: In cancer, cPRC1 complexes have a dual function. cPRC1 is recruited to gene promoters to repress gene expression and to active cancer-specific enhancers in different cancer subtypes to modulate their expression and chromatin accessibility to oncogenic transcription factors

these results confirm that RING1B has dual function in regulating transcriptional programs in breast cancer cells and does so by altering chromatin accessibility for key transcription factors and chromatin organization proteins.

**RING1B is recruited to enhancers in other cancer types**. We finally sought to determine whether RING1B recruitment to SEs only occurs in breast cancer cells or if, in contrast, RING1B acquired the ability to bind to enhancers in other cancer types. To this end, we used public RING1B ChIP-seq data sets from ENCODE in a leukemia cell line, K562, and in a hepatocellular carcinoma cell line, HepG2. Notably, in both cell lines, RING1B co-localized with the enhancer-associated histone modifications (Supplementary Fig. 9a). We identified 1246 SEs in HepG2 and 852 SEs in K562 cells, of which 66 and 95% contain RING1B, respectively (Fig. 7a, b). Moreover, RING1B peaks that co-localized with H3K4me1 and H3K27ac were devoid of H3K27me3 (Supplementary Fig. 9b) and about 40% of typical

enhancers were occupied by RING1B in both HepG2 and K562 cells (Supplementary Fig. 9c).

RING1B-containing SEs in breast cancer cells included binding sites for oncogenic transcription factors (Fig. 2g). In HepG2 cells, RING1B was recruited to SEs bound by key circadian rhythm transcription factors, including CLOCK, BMAL, and NPAS2, that directly regulate the expression of BHLHE40, another core clock component identified in our analysis (Fig. 7c)[37]. Importantly, disruption of the circadian clock has been implicated in liver cancer and the abnormal expression of clock genes correlates with increased tumor size and cell proliferation[38,39]. Moreover, using the TCGA liver hepatocellular carcinoma (LIHC) data set, we found that 9% of patients with LIHC have amplification of *RNF2* (Supplementary Fig. 9d), and its expression is significantly higher in liver tumors compared to normal liver (Supplementary Fig. 9e). Finally, we confirmed that RING1B-containing enhancers in HepG2 cells were also decorated with BHLHE40 (Fig. 7d, e), and 72% of SEs contained both RING1B and BHLHE40 (Supplementary Fig. 9f, left).

In K562 cells, RING1B was recruited to SEs containing binding motifs for GATA1 and GATA2 factors (Fig. 7f). GATA2 is often mutated in myeloid malignancies while GATA1 is overexpressed in acute myeloid leukemia (AML), highlighting the role of GATA factors in leukemia[40]. Importantly, we also confirmed that GATA1 is recruited to RING1B-containing enhancers and to 65% of the SEs in K562 cells (Fig. 7g, h and Supplementary Fig. 9f, right). As expected, RING1B binding to SEs in K562 and HepG2 cells was cell type specific (Fig. 7i). These observations lead us to conclude that RING1B is a novel epigenetic factor that promotes important transcriptional regulatory networks at enhancers to promote oncogenic pathways in multiple cancer types (Fig. 7j).

## Discussion
During the last decade, extensive sequencing of cancer genomes has revealed mutations of transcription factors, epigenetic machineries, and signaling pathway factors, and led to the development of novel therapeutic targets. Nonetheless, it remains crucial to investigate the molecular mechanisms, enzymes, and epigenetic machineries that are dysregulated and altered in cancer. The importance of this complementary approach is exemplified by the PRC1-mediated mechanisms we propose. While PRC1-encoding genes are not typically mutated in cancer, we found that several canonical *PRC1* genes are amplified and dysregulated in many hormone-related cancers, including breast cancer. Hormone-related cancers share a unique mechanism of action, as hormones drive proliferation which induces the accumulation of mutations[41]. Whether hormones contribute to chromosome instability and genomic rearrangements of genomic sites of *PRC1* genes in breast cancer remain to be addressed.

We propose that in normal breast epithelial and breast cancer cells, RING1B function is uncoupled from its classical role as a repressor of lineage genes[2]. In pluripotent cells, RING1B is the main E3-ligase that mono-ubiquitinates histone H2A[42]. In contrast, in the breast cancer cells used in this study, RING1A is more enzymatically active towards histone H2A than RING1B. TRIM37 was recently proposed as a novel histone H2A ubiquitin ligase in breast cancer cells, with a chromosomal copy-number amplification at 17q23[43]. T47D and MDA-MB-231 cells do not have amplification of this chromosome arm, which is consistent with our results suggesting that RING1A is the main histone H2A mono-ubiquitin ligase in these cells. Further analyses at the biochemical level are required to determine the exact mechanisms underlying RING1A and TRIM37 deposition of H2AK119ub1 in the context of breast cancer.

Although the classical PRC1-mediated gene regulation is to compact chromatin, PRC1 complexes are also involved in facilitating gene transcription[6,7,28,44,45]. The exact function of PRC1 complexes in gene activation, and the molecular mechanisms that permit Polycomb complexes to activate genes, are under intense investigation. Recently, it has been shown that the PRC1 complex is redistributed genome-wide during oncogenesis in Drosophila to sites decorated with H3K27ac[7]. In melanoma, RING1B is recruited to chromatin to repress gene activity, but it is also recruited to transcriptionally active genes devoid of H3K27me3 and H2AK119ub1[44]. More recently, it has been shown than in adult epidermal stem cells, PRC1 is recruited to gene promoters containing histone modifications typically found in active enhancers[45]. Here, we have provided the first evidence that RING1B and cPRC1 complexes are recruited to enhancer regions in cancer cells and that RING1B depletion has a major impact on chromatin accessibility at enhancers. Mechanistically, we show a functional crosstalk between RING1B and the FOXA1/ERα axis, which ultimately resulted in an attenuated response to estrogen. FOXA1, as a nuclear receptor regulatory factor, is not limited to ERα as it also interacts with the androgen receptor (AR) to regulate its deposition to chromatin in prostate cancer cells[46]. Thus, it remains to be determined whether RING1B functionally associates with FOXA1 and AR in other cancer types or during embryonic development.

None of the cPRC1 components have the ability to directly bind DNA. Therefore, we anticipate that non-genomic factors are involved in their recruitment to chromatin. One possibility could be that cPRC1 components are post-transcriptionally modified at SEs. Since CBX proteins can bind RNA[47], another possible mechanism could involve CBX8 interacting with eRNAs to recruit RING1B and the cPRC1 complex to specific enhancers. Interestingly, physical interactions between PRC1 and Fs(1)h and Br140, the *Drosophila* orthologs of BRD4 and BRD1, have been recently described[48]. Although we did not detect a physical interaction of RING1B with BRD4 in the MDA-MB-231 cell line under our experimental conditions, we observed a strong co-occupancy of BRD4 and cPRC1 at active genes, in agreement with the observation of Kuroda and colleagues. Our studies also indicate that both of these epigenetic machineries are co-recruited to enhancer regions in breast cancer cells.

Recruitment of RING1B to active enhancers would appear to suggest that RING1B is solely involved in positively regulating their expression. Surprisingly, we discovered a much more complicated scenario, in which RING1B can exert canonical and non-canonical functions at enhancers, both at the levels of their transcriptional activity and transcription factor accessibility. We theorize that in a set of highly active enhancers, RING1B is required for their activity, while in another set of enhancers with diminished activity, RING1B is required to prevent a hyperactivation of the enhancer. This model is in agreement with a recent report showing that RACK7 and KDM5C are recruited to enhancers, where they act to hamper full enhancer activation in cancer cells[49]. Overall, we suggest that intricate epigenetic mechanisms mediate enhancer activity and disruption of this regulation may contribute to tumorigenesis.

The contribution of Polycomb complexes in breast cancer tumorigenesis and metastasis is largely unknown. Here we show that high levels of *RNF2* in patients with ER+ breast cancer tumors correlate with good survival outcome, while high *RNF2* levels in patients with basal breast cancer correlate with lower survival probability. These surprising results are consistent with our xenografts experiments and with other reports showing that RING1B is required for migration of TNBC cells[50]. It has been also shown that high levels of RING1B correlated with metastatic squamous cell carcinoma[50,51]. The molecular mechanisms by

which RING1B either prevents or enhances metastasis in specific breast cancer subtypes remain to be fully understood. Future work coupling genomics and genome architecture with functional assays may help reveal which of the RING1B-mediated molecular mechanisms contribute to breast cancer metastasis. Finally, we propose that development of small molecules to impair RING1B recruitment to specific genomic sites in TNBC tumors may have important therapeutic implications.

## Methods

**Cell lines.** Human iPSCs (ATCC #ACS-1021) were maintained in complete feeder-free mTESR1 culture medium (STEMCELL Technologies #85850) on matrigel-coated plates (Corning #354277) at 37 °C with 5% $CO_2$. The culture medium was changed daily and iPSCs colonies were enzymatically passaged with StemPro Accutase Cell Dissociation Reagent (Thermo Fisher Scientific #A1110501) at a 1:4–1:6 split ratio every 4–7 days. DMEM/F-12 media (STEMCELL Technologies #36254) was used to detach colonies. ROCK inhibitor Y-27632 (STEMCELL Technologies #72302) was used in every split and when cells were thawed from liquid nitrogen. If identified, spontaneously differentiated cells were mechanically removed prior to passaging. MCF10A, MDA-MB-231, T47D, and SKRB3 (ATCC #CRL-10317, HTB-26, HTB-133, and HTB-30) were maintained at 37 °C with 5% $CO_2$ and split every 2–3 days according to ATCC recommendations. Culture media was supplemented with 1× penicillin/streptomycin (Thermo Fisher Scientific #15140-122) and 1× glutamax (Thermo Fisher Scientific #35050-061), and complete culture media for each cell line were as follows: MCF10A—DMEM/Ham's F-12 (1:1) (Corning #45000-348) with 5% horse serum (Thermo Fisher Scientific #16050-122), 10 ng/ml EGF (Thermo Fisher Scientific #PHG0311), 50 ng/ml cholera toxin (Sigma-Aldrich #C8052), 10 µg/ml insulin (Sigma-Aldrich #91077 C), and 500 ng/ml hydrocortisone (Sigma-Aldrich #H0888); MDA-MB-231—DMEM (Lonza #12001-576) with 10% FBS (Benchmark #100-106); T47D—RPMI-1640 (Lonza #95042-506) with 10% FBS and 10 µg/ml insulin; SKBR3—McCoy's 5a Medium Modified (Lonza #12001-562) with 10% FBS. For experiments in which estrogen (10 nM E2) was added to T47D, cells were maintained in phenol-free RPMI-1640 (Thermo Fisher Scientific #32404014) supplemented with 5% charcoal-stripped FBS (Benchmark #100-119) for 72 h prior to treatment. In experiments with BRD4 inhibition, MDA-MB-231 cells were treated either with DMSO or 500 nM JQ1, obtained from the Bradner lab. Cells were routinely tested to be free of mycoplasma infection. Cells were imaged using an Olympus IX70 inverted microscope fitted with a phase-contrast filter.

**Generation of cells stably expressing shRNAs.** To produce shRNA lentiviruses, $2 \times 10^6$ HEK293T cells (ATCC #CRL-3216) were plated into a 10 $cm^2$ plate and transfected 16 h later with 8 µg of pLKO-shRNAs (Addgene #10879 for CTR; Sigma-Aldrich #TRCN0000033696 and TRCN0000033697 for RING1B; and Sigma-Aldrich #TRCN0000014881 for FOXA1), 2 µg of pCMV-VSV-G, and 6 µg of pCMV-dR8.91 plasmids using calcium phosphate. 72 h after transfection, viral supernatant was collected, passed through a 0.45 µM polyethersulfone filter, and used to transduce MDA-MB-231 and T47D cells. Specifically, $3 \times 10^5$ cells were plated into a 6-well plate followed by the addition of viral media with 8 µg/ml polybrene (Millipore-Sigma #TR-1003-G). Cells were centrifuged for 1 h at 1000×$g$ at 32 °C then incubated overnight with fresh viral media. Viruses were removed and complete culture media was added for cell recovery. Cells were selected 24 h after recovery with 2 µg/ml of puromycin (Biogems #5855822) and were maintained in selection. All experiments were performed within 3 weeks post transduction.

**Western blotting and immunoprecipitation.** Cells were lysed in high-salt buffer (300 nM NaCl, 50 mM Tris-HCl [pH 8], 10% glycerol, and 0.2% NP-40) supplemented with protease inhibitors (Sigma-Aldrich #04693132001) and sonicated 5 min at 4 °C with a Bioruptor in 30″ ON-OFF cycles. After centrifugation at 16,000×$g$ for 15 min, soluble material was quantified by Bradford assay (Bio-rad #5000006), and 1 mg of protein was used for each immunoprecipitation (IP), or 30–50 µg of protein was loaded onto SDS-PAGE gels for western blotting. IP samples were incubated overnight with 5 µg of antibody (see Supplementary Data 6 for a list of antibodies used) followed by 30 µl of protein A/G agarose bead slurry (Santa Cruz #sc-2003) for 2 h. IP material was washed 3× with high-salt buffer and eluted with Laemmli buffer, then loaded for SDS-PAGE. Western blotting was performed using standard protocols and imaged on an Odyssey CLx imaging system (Li-COR), and various exposures within the linear range captured using ImageStudio software (Li-COR). Images were rotated, resized, and cropped using Adobe Photoshop CC 2018 and imported into Adobe Illustrator CC 2018 to be assembled into figures. Unprocessed images for all western blots in main figures are provided in Supplementary Fig. 10.

**Subcellular fractionation.** Two 150 $cm^2$ dishes with 80% confluent T47D and MDA-MB-231 cells (control, RING1B or FOXA1-depleted cells) were used for each experiment. All steps were carried out at 4 °C. Cells were collected with a cell

scraper and washed 1× with PBS, then centrifuged for 5 min at 400×$g$. Cell pellet was resuspended 1:5 (w:v) in Buffer A (10 mM HEPES [pH 7.9], 1.5 mM $MgCl_2$, 10 mM KCl, 0.05% NP-40) supplemented with protease inhibitors and 0.5 mM DTT. After 10 min on ice, cells were centrifuged for 5 min at 400×$g$. The supernatant, representing the cytosolic fraction, was collected and stored at 4 °C. The remaining nuclear pellet was resuspended in ¾ of the initial volume with Buffer B (5 mM HEPES [pH 7.9], 1.5 mM $MgCl_2$, 0.2 mM EDTA, 26% glycerol) supplemented with protease inhibitors and 0.5 mM DTT, and homogenized with 20 strokes in a dounce homogenizer fitted with pestle A. After 20 min on ice, extracts were centrifuged for 20 min at 16,000×$g$. The supernatant, representing the soluble nuclei fraction, was collected and stored at 4 °C. The remaining pellet was resuspended in ½ of the original volume with high-salt buffer (300 mM NaCl, 50 mM Tris-HCl [pH 8], 10% glycerol, and 0.2% NP-40) supplemented with protease inhibitors and sonicated with a Bioruptor for 5 min in 30″ ON-OFF cycles. After centrifugation at 16,000×$g$ for 20 min, the supernatant, representing the soluble chromatin fraction, was stored at 4 °C. The remaining pellet, representing the insoluble chromatin fraction, was resuspended in volume equal to original volume with 2×Laemmli buffer and sonicated with a Bioruptor for 10 min in 30″ ON-OFF cycles. Protein concentration of cytosolic, soluble nuclei and soluble chromatin fractions were determined by Bradford assay, and 20 µg of protein and 1/5 of the insoluble chromatin fraction material were loaded onto SDS-PAGE gels followed by western blotting.

**Transfection of siRNAs.** The day before siRNA transfection, $2 \times 10^5$ cells were seeded in 6-well plates and maintained in antibiotic-free medium. 25 nM siRNAs (Sigma-Aldrich #SIC007 for CTR, EHU230291 for RING1A, EHU109061 for RING1B, and EHU155811 for FOXA1) were transfected using Lipofectamine RNAiMAX (Thermo Fisher Scientific #13778150) following the manufacturer's instructions.

**Animal studies.** The University of Miami Institutional Animal Care and Use Committee (IACUC) approved all animal procedures. shCTR and shRING1B T47D and MDA-MB-231 cells were transduced with retroviruses expressing GFP-luciferase (pMSCV-IRES-Luciferase-GFP), and successful transduction confirmed by imaging cells on an Olympus IX70 fluorescence microscope with a GFP filter. After transduction of cells, GFP-positive cells were collected by FACS. $1 \times 10^6$ shCTR and shRING1B-MDA-MD-231$^{GFP-luc}$ cells and $5 \times 10^6$ shCTR and shRING1B T47D$^{GFP-luc}$ cells were injected into the mammary fat pad of 8-week-old female NSG mice (Jackson Labs #005557) ($n = 5$ per group). Sample size was chosen to generate enough power for statistical significance and mice were randomly allocated to experimental groups, estimating variance is similar for the two groups. Injection of tumor cells were not blinded. Tumor development and metastasis were monitored weekly using an in vivo imaging system (IVIS, Perkin Elmer) during the course of 65 and 72 days for MDA-MB-231 cells and TD47, respectively. Specifically, 10 min prior to imaging, mice were injected intraperitoneally with D-luciferin (Perkin Elmer #760504) at a dose of 150 mg/kg. Tumor size and metastasis were quantified using the Living Image software (Perkin Elmer). Luciferase signal is represented as Luminescence (Photons/s). Mice were sacrificed at the indicated time points and primary tumors were collected and weighed. No animals were excluded from the analysis.

**Preparation of ATAC reactions and libraries.** ATAC-seq experiments were performed as previously described[35] with modifications. Briefly, 25,000 cells of each cell line were used to perform the transposition reaction. Samples were eluted in 13 µl of Buffer EB (Qiagen #28206). To calculate the number of cycles for library amplification, 2 µl of transposed DNA were amplified by qPCR for a total of 25 cycles. The 10 µl qPCR reaction was set up as follows: 2 µl of transposed DNA, 0.3 µl 25 µM Ad1_noMX, 0.3 µl 25 µM Ad2.X (custom oligos synthesized by Integrated DNA Technologies, see Supplementary Data 7), 5 µl NEBNext High-Fidelity 2X PCR Master Mix (New England BioLabs #M0541S), 0.1 µl 100X SYBR Green I (Thermo Fisher Scientific #S7563) and 2.3 µl nuclease-free water with the following program on a Bio-Rad CFX96 Optics Module Thermal Cycler machine: (1) 72 °C for 5 min, (2) 98 °C for 30 s, (3) 98 °C for 10 s, 63 °C for 30 s and 72 °C for 30 s, 25 cycles, (4) 72 °C for 1 min, and (5) hold at 10 °C. The Ct value of each sample reflects the number of PCR cycles for optimal amplification in the linear range of the reaction. A 50 µl PCR reaction was then set up as follows: 10 µl transposed DNA, 1.5 µl 25 µM Ad1_noMX, 1.5 µl 25 µM Ad2.X (unique for each sample), 12 µl nuclease-free water, and 25 µl NEBNext High-Fidelity 2X PCR Master Mix with the same program as for the qPCR, but substituting the cycle number with the Ct-value obtained from the qPCR reaction. The PCR was performed on a Bio-Rad C1000 Touch Thermal Cycler. After PCR, the 50 µl reactions were cleaned up and size selected by adding 25 µl AMPure XP beads (Beckman Coulter #A63881) to remove fragments higher than 800 bp. The supernatant was transferred to a new tube and 65 µl AMPure XP beads were added to remove fragments smaller than 100 bp, then washed twice with freshly prepared 80% ethanol and eluted in 25 µl nuclease-free water. To determine the average fragment size of each library, samples were run through a high sensitivity DNA screentape (Agilent Technologies #5067–5584) following the manufacturer's instructions on an Agilent Technologies 2200 TapeStation machine. To determine the

concentration of each library, Qubit dsDNA high sensitivity reagents (Thermo Fisher Scientific #Q32851) were used following the manufacturer's instructions on a Qubit 3 fluorometer. Finally, the samples were pooled and sequenced, paired-end, 75 bp on a NextSeq 500.

**ATAC-seq analysis.** FASTQ data were processed with Trimmomatic v0.32 and Skewer v0.2.2 to remove low-quality reads, and paired-end reads were aligned to the hg19 genome (UCSC) using Bowtie2 v2.2.6. Duplicate reads were removed using Picard tools (version 1.126 -http://broadinstitute.github.io/picard). Read alignment was offset as previously described[35]. Peaks were called using the MACS2 v2.1.0.20150731 algorithm with the parameters: -g hs -p 0.01—nomodel—shift −75—extsize 150 and a cutoff of $q$-value < 0.05. Bedtools v2.26.0 intersect was used to determine peak overlaps. NGS Plot was used to generate heat maps and density plots. Homer annotatePeaks was used for peak annotation.

**Purification of endogenous RING1B complexes.** Cell pellets in triplicate from MDA-MB-231 or T47D cell lines were snap frozen in liquid nitrogen. Each replicate was processed as follows, modified from established procedures[52]. Each pellet was resuspended 1:4 (w:v) in a solution of 50 mM Tris pH 8.0, 300 mM NaCl, 0.2% (v/v) NP-40, supplemented with protease inhibitors (Sigma-Aldrich #11836170001). Samples were lysed by ultrasonication at 4 °C using a QSonica S4000 equipped with an S4717 microtip probe. For each sample, 2-s-long pulses at 1Amp were applied, with 1 s pauses, until ~20 J of output per 100 mg of cell pellet was reached. After sonication, samples were centrifuged at 20,000×$g$ at 4 °C for 10 min, producing a clarified cell extract. For MDA-MB-231, 400 µl of clarified extract from each replicate was used in IP and control experiments, respectively. In IP experiments, extracts were combined with 10 µl of magnetic affinity medium (Thermo Fisher Scientific #14301) coupled to anti-RING1B antibodies (MBL, see Supplementary Data 6). In control experiments mouse IgG (Sigma-Aldrich, see Supplementary Data 6) was used. 7.3 µg of anti-RING1B antibody and 10 µg of mouse IgG were used, respectively, per mg of magnetic medium in epoxy-based covalent coupling (as per manufacturer's instructions). For T47D IP/control experiments, 200 µl of extract were combined with 5 µl of affinity medium. Clarified cell extracts were incubated with magnetic media for 1 h at 4 °C with gentle end-over-end mixing. After mixing, the supernatants were removed and the beads were washed three times with 1 ml of the extraction solution. The bound fraction was released from IP/control experiments by the addition of 15 µl of 1× LDS sample buffer (Thermo Fisher Scientific #NP0007) with incubation at 70 °C for 5 min with agitation. After incubation, the eluate was removed, reduced with DTT, alkylated with iodoacetamide, and then run a ~6 mm into a 4–12% Bis-Tris NuPAGE gel (1 mm, 12-well; Thermo Fisher Scientific #NP0322BOX). The gel was stained with Coomassie Brilliant Blue G-250, the samples (gel plugs) were excised, and cut into ~1 mm cubes for processing, as follows. Samples were destained with several washes of 0.5–1 ml 50% v/v acetonitrile (ACN) in 50 mM ammonium bicarbonate at 37 °C with shaking. Destained gel pieces were dehydrated by washing with 500 µl ACN, and placed in a speed-vac for ~10 min at RT. Trypsin working solution (~40 µl at 12.5 ng/µl in 50 mM ammonium bicarbonate) was added to the gel pieces on ice, which were allowed to swell for 45 min. After swelling, additional 50 mM ammonium bicarbonate was added in order to submerge the swollen gel pieces (typically ~15 µl). The samples were incubated at 37 °C to undergo tryptic proteolysis. Trifluoroacetic acid (TFA) was added to each tube at 0.5% (w/v) final concentration and incubated 5 min at RT. The supernatant (tryptic digest supernatant) was recovered and transferred to a 0.5 ml low protein-binding microfuge tube (Sorenson Bioscience #11300). An aliquot of 50 µl 0.1% w/v TFA was added to the gel pieces, which were extracted a further 45 min at RT, with agitation. The supernatants were removed and pooled with the appropriate tryptic digest supernatant. Pooled extracted peptides were desalted using C18 reversed-phase OMIX tips (Agilent #A57003100) as per the manufacturer's instructions. The peptides were eluted from the tips first with 100 µl of aqueous 40% (v/v) ACN, 0.5% (v/v) acetic acid (E1) and then with 100 µl of 80% (v/v) ACN, 0.5% (v/v) acetic acid (E2). E1 and E2 were combined, frozen in liquid nitrogen, and dried in a centrifugal vacuum concentrator.

**Strand-specific total RNA library preparation.** RNA was isolated from fresh or frozen cell pellets using TRIzol reagent (Thermo Fisher Scientific #15596018). Ribosomal RNA was removed using the NEBNext rRNA Depletion Kit (New England Biolabs #E6310) starting with 1 µg total RNA and following the manufacturer's instructions. Ribosomal RNA-depleted samples were then further processed with the NEBNext Ultra Directional RNA Library Prep Kit for Illumina (New England Biolabs #E7420) for library preparation following the manufacturer's instructions. Libraries were pooled and sequenced, single-end, 75 bp on a NextSeq 500.

**ChIP and ChIP-seq library preparation.** Cells were grown to 80% confluence on 150 cm$^2$ plates and processed for ChIP of histone modifications or ChIP of non-histone targets. For histone modifications, cells were fixed with 1% formaldehyde (Sigma-Aldrich #252549) added directly to culture media for 15 min, shaking gently at RT. During crosslinking, 1.25 M glycine solution (10X stock) was prepared, then added to plates at a final concentration of 125 mM for 5 min, shaking

gently at RT. The supernatant was aspirated, cells washed 1× with PBS, and harvested on ice using cell scrapers into 15 ml sonication tubes (Diagenode #C01020031). Cells were pelleted at 400×$g$ for 3 min at 4 °C, washed 1× with 10 ml cold PBS, and pelleted once more. Cell pellet was resuspended in 1.3 ml cold ChIP Buffer (two volumes of SDS ChIP Buffer [100 mM NaCl, 50 mM Tris-HCl pH 8.1, 5 mM EDTA pH 8.0, 0.5% SDS] with one volume TXT ChIP Buffer [100 mM Tris-HCl pH 8.6, 100 mM NaCl, 5 mM EDTA pH 8.0, 5% Triton X-100]). Cells were then sonicated with a Bioruptor Pico at 4 °C for 10 min of 30" ON-OFF cycles for iPSCs or 20 min for MCF10A, MDA-MB-231 and T47D. After sonication, samples were centrifuged at 16,000×$g$ for 15 min at 4 °C and supernatant transferred to a new 1.5 ml microtube. To check sonication efficiency, 20 µl of sonicated samples was transferred to a new microtube with 80 µl PBS and incubated at 65 °C for 3 h on an Eppendorf Thermomixer shaking at 1000 rpm to decrosslink. DNA was purified with the QIAquick PCR Purification Kit (Qiagen #28106) and eluted in 30 µl H$_2$O. 6 µl of orange DNA dye was added and 12 µl and 24 µl of each sample were run in a 1% agarose gel at 100 V for 15 min and imaged on a Bio-rad ChemiDoc XRS + . Protein concentration in sonicated samples was measured by Bradford assay, and 200 µg of total protein was transferred to a 1.5 ml LoBind tube (Eppendorf #0030108051) and brought up to 500 µl final volume with ChIP buffer. 5 µl was removed as input material (1%) and placed in a separate microtube at 4 °C. 2 µg antibody was used for each histone ChIP, except for H2AK119ub in which 1.5 µg was used (see Supplementary Data 6). The samples were rotated end-to-end overnight at 4 °C. Protein A sepharose beads (GE Healthcare #17528001) were washed 3× with ChIP buffer and 30 µl bead slurry was added to each sample. Samples were incubated with beads for 2 h at 4 °C rotating end-to-end. Following incubation, samples were centrifuged at 400×$g$ for 3 min at 4 °C, washed 2× with ChIP Low Salt Buffer (50 mM HEPES pH7.5, 140 mM NaCl, 1% Triton X-100, 1x protease inhibitors), and 1× with ChIP High-Salt Buffer (50 mM HEPES pH7.5, 500 mM NaCl, 1% Triton X-100, 1× protease inhibitors). Beads were dried after the last wash with a 28 G needle fitted to a 1 ml syringe. Elution Buffer (1% SDS, 0.1 M sodium carbonate) was prepared fresh before use and 110 µl was added to each sample and 95 µl to input sample that was previously set aside. To elute immunocomplexes from beads, samples were incubated at 65 °C for 3 h on an Eppendorf Thermomixer shaking at 1000 rpm. Tubes were centrifuged for 3 min at 400×$g$ at RT and 100 µl of supernatant was transferred to a new tube, being careful not to aspirate beads. DNA purification was performed with the QIAquick PCR Purification Kit (Qiagen cat# 28106) and eluted in 60 µl H$_2$O and quantified by Qubit. For non-histone ChIP targets, samples were processed using the High Sensitivity ChIP-IT Kit (Active Motif #53040, see Supplementary Data 6). Immunoprecipitated DNA from both methods were used to either perform qPCR or generate libraries using the NEBNext Ultra DNA Library Prep Kit for Illumina (New England Biolabs #E7370) following the manufacturer's instructions. Libraries were visualized on a Tapestation 2200 using D1000 DNA screentape (Agilent Technologies #5067–5582). Libraries were quantified on a Qubit 3 fluorometer with Qubit dsDNA high sensitivity reagents (Thermo Fisher Scientific #Q32851) following the manufacturer's instructions, then pooled and sequenced, single-end, 75 bp on a NextSeq 500. Processed data was viewed using the UCSC genome browser with a smoothing window of 5 pixels. ChIP-qPCR was performed using primers targeting developmental or enhancer regions identified (see Supplementary Data 8 for list of primers) on a Bio-Rad CFX96 Real-Time System with iTaq universal SYBR green supermix (Bio-rad #1725124) and analyzed with CFX Manager software (Bio-Rad).

**TCGA data preparation and analysis.** The legacy level 3 data of breast invasive carcinoma (BRCA) and liver hepatocellular carcinoma (LIHC) from the The Cancer Genome Atlas (TCGA) cohort were obtained from the Genomic Data Commons (GDC) data portal. RNA-seq raw counts of 1211 BRCA and 421 LIHC cases as legacy level 3, and using the hg19 human reference genome, were downloaded, normalized and filtered using the R/Bioconductor package, TCGA-biolinks version 2.5.9. GDCquery, GDCdownload and GDCprepare were used for both tumor types ("BRCA" and "LIHC", level 3, and platform "IlluminaHi-Seq_RNASeqV2"). Integrative analysis using mutation, clinical classification, and gene expression were performed following our recent TCGA's workflow[53]. Among BRCA samples 1097 were Primary Solid Tumor (TP) and 114 were solid Tissue Normal (NT). Among LIHC samples 371 were TP and 50 NT. The aggregation of the two matrices (tumor and normal) for both tumor types was then normalized using within-lane normalization to adjust for GC-content effect on read counts and upper-quantile between-lane normalization for distributional differences between lanes, applying the TCGAanalyze_Normalization function and adopting the EDASeq protocol. Molecular subtypes, mutation data, and clinical data were pulled using TCGAbiolinks and the following functions: TCGAquery_subtype, GDCquery_maf retrieving somatic variants that were called by the MuTect2 pipeline, and GDCquery_clinic. BRCA tumors with PAM50 classifi-cation[54] were stratified into five molecular subtypes: Basal-like ($n = 98$), HER2-enriched ($n = 58$), Luminal A ($n = 231$), Luminal B ($n = 127$), and Normal-like ($n = 8$). Normal-like samples were not considered in this analysis due for the limited number of sample availability. For LIHC, tumors with iCluster classifica-tion were stratified into three molecular subtypes: iCluster:1 ($n = 65$), iCluster:2 ($n = 55$), and iCluster3 ($n = 63$). Tumor stage information was retrieved from the clinical data grouping to main stages (I, II, III, IV) and each subgroup (Ia, IIb, IIIc

etc.). Amplification data obtained from the GISTIC 2.0 tool was then used to identify regions of the genome that were significantly amplified or deleted across a set of samples[55]. GISTIC2 data was retrieved from cBioPortal for both tumor types considering samples with high amplification greater than 2 and excluding high deletion samples lower than −2. The ggpubr R/CRAN (https://CRAN.R-project.org/package=ggpubr) package was used to draw box plots showing relative expression for each cancer type, stage, and molecular subtype and to perform multiple means comparisons using a non-parametric Wilcox test. All analyses and plots were generated using the R environment (see Supplementary Data 9 for list of software).

**ChIP-Seq analysis**. ChIP-seq of RING1B, H3K27ac, H3K27me3, and H3K4me1 (in K562 and HepG2 cells), GATA1 (in K562 cells) and bHLHE40 (in HepG2 cells) were re-analyzed from ENCODE data sets (GSE95908, GSE91837, GSM733658, GSM733754, GSM733656, GSM733743, GSM733692, GSM798321, GSM1003608, GSM935566, GSM733780, GSM733732). All ChIP-seq data, generated in this study or deposited into ENCODE, were analyzed according to the following methodology: FASTQ data were processed with Trimmomatic v0.32 to remove low-quality reads and then aligned to the human genome hg19 using BWA v0.7.13-r1126 with the following parameters: aln -q 5 -l 32 -k 2. Duplicate reads were removed using Picard tools (version 1.126—http://broadinstitute.github.io/picard). Peaks were called using MACS2.1 with default parameters –shiftsize 160 –nomodel –p 0.01 for all data except RING1B in human iPSCs. For all H3k27me3 and H2AK119ub1 peak calling, the option –broad was added to identify broad peaks. Whole-cell extract input from the corresponding cell lines were used as controls. Peaks with fold change > 4 and a $q$-value < 0.05 were used for downstream analysis. Bigwig file output from MACS v 2.1.0.20150731 was visualized in the UCSC genome browser. Homer annotatePeaks v4.8.3 was used for peak annotation. Intergenic peaks are located greater than −2.5 kb from the TSS of a gene; intragenic peaks are inside genes, including intron, exons, and UTRs; −2.5 kb + TSS peaks are located at the TSS and a maximum of 2.5 kb upstream of the TSS; and other peaks are located at uRNAs, microRNAs, and pseudogenes. Bedtools v2.26.0 intersect was used to determine peak overlaps. NGS Plot v2.61 was used to generate heat maps and density plots (see Supplementary Data 9 for list of software).

**Identification of super-enhancers**. Super-enhancers (SE) and typical enhancers were defined using the ROSE pipeline (Supplementary Data 9) with default parameters using H3K27ac ChIP-seq peaks as input. Only expressed protein coding genes were considered as potential SE targets. A gene was considered expressed if FPKM > 1 in both replicates of RNA-seq from the same condition. Expressed genes in shCTR and shRING1B were compiled into one and Bedtools closestBed was used to determine the closest expressed genes to SE regions, with the parameter −D to report both upstream and downstream genes. The output was filtered to include only genes ±200 kb flanking both sides of the SE.

**Mass spectrometry analysis**. For MS analysis, dried peptide samples were resuspended in 10 μl of aqueous 5% (v/v) methanol, 0.2% (v/v) formic acid. Mass spectra were recorded on a QExactive Plus mass spectrometer (Thermo Fisher Scientific). Database searching and label-free quantitation were performed by MaxQuant using the UP000000589 human database. Intensities were based on maximum peak height. The "proteingroups.txt" file was uploaded to Perseus, and protein identifications from the decoy database were removed. LFQ intensities were logarithmized. Control experiments were grouped together, as were RING1B pull-down experiments. Proteins were filtered, with the constraint that at least one group (RING1B or control) should contain at least three valid values. Missing values were imputed from a normal distribution. Relative protein abundance was calculated by normalizing the LFQ intensity by protein mass. A two-sample Student's $t$-test was performed with a permutation-based FDR = 0.01 used for truncation.

**RNA-seq analysis**. FASTQ data were processed with Trimmomatic v0.32 to remove low-quality reads and then aligned to the human genome hg19 using STAR aligner (version 2.5.3a) with default parameters and RSEM (version 1.2.31) to obtain expected gene counts against the human RefSeq (release 76). Differential expression was determined between RING1B (*RNF2*) shRNA and scrambled shRNA using DESeq2 and R (version 3.2.3) with $q$-value < 0.05. Heat maps were generated using SpotFire with Decision Site for Functional Genomics (SpotFire Inc., Somerville, MA, USA).

**Gene ontology analysis and gene set enrichment analysis**. The gene ontology (GO) analysis and pathway enrichment analysis were performed with EnrichR (2016 update) using the differentially expressed genes or the genes closest to RING1B peaks. Gene expression in fold changes was obtained as described above, and the entire list of expressed genes was pre-ranked and imported into the gene set enrichment analysis (GSEA).

**Genome-wide identification of eRNA loci**. For eRNA identification, the BEDtools window function was used to overlap H3K27ac and H3K4me1 peaks in a window

of ±200 bp. Next, BEDtools intersect was used with the option –v to discard any peak overlapping any exons from the hg19 reference genome (Gencode version 27), with additional 2 kb surrounding every exon. A ±600bp window at the center of the H3K27ac peak was used to calculate RPKM across the entire eRNA locus using total RNA-seq data, considering regions with eRNA expression to have RPKM > 0.3. Next a cutoff of fold change >2 or <−2 (shCTR versus shRING1B) was used to detected differential expression. eRNA loci was overlapped with super-enhancer regions or typical enhancer regions using BEDtools intersect.

**Motif analysis**. Motif finding was performed with Homer findMotifs v4.8.3. A window of ±100 bp (option -d 200) relative to peak summits was used to perform the analyses.

**Statistical analysis**. Significance was determined by either Student's $t$-test, non-parametric Wilcox test, Mann–Whitney test, or Kolmogorov–Smirnov test, as indicated. Error bars in figures represent standard deviation (SD) of at least two independent experiments.

**Data availability**. All of the genome-wide data of this study have been deposited in the NCBI Gene Expression Omnibus (GEO) database, GSE number: GSE107176. The mass spectrometry proteomics data have been deposited to the ProteomeXchange Consortium via the PRIDE partner repository with the data set identifier PXD009570.

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

## Acknowledgements

We are indebted to V.A. Raker for help in preparing the manuscript, to Gloria Mas Martin and Dr. Joyce M. Slingerland for discussions, and to the Oncogenomics Core Facility at the Sylvester Comprehensive Cancer Center for performing high-throughput sequencing. We also thank the Flow Cytometry Core Facility for assistance with cell sorting, and the Cancer Modeling Shared Resource for assistance with the animal studies. Dr. Ramiro E. Verdun kindly provided the shRING1B plasmids. Kelly Molloy assisted with the mass spectrometry. Dr. Derek M. Dykxhoorn assisted with the iPSC culture. Dr. Aznar-Benitah provided the pMSCV-IRES-Luciferase-GFP vector. This work was supported by Sylvester Comprehensive Cancer Center funds to L.M.

## Author contributions

L.M and H.L.C designed the study. H.L.C conducted all the experiments, except ATAC-seq (J.G.H in the laboratory of R.S and L.M), siRNA and cellular fractionation experiments (Y.Z), and the RING1B pull-downs and LC-MS/MS experiments (H.J under the supervision of J.L). Bioinformatics analyses were performed by F.B (in the laboratory of R.S.) and H.L.C TCGA analysis was performed by A.C (in the laboratory of M.E.F). D.B is the Head of the Cancer Modeling Shared Resource Facility at the Sylvester Comprehensive Cancer Center and performed the animal studies. L.M supervised the experiments, performed the experiments in iPSCs and provided intellectual support toward interpretation of the results. L.M and H.L.C wrote the manuscript.

## Additional information

**Competing interests:** The authors declare no competing interests.

