## [Peer Review File · Nature Communications]

Reviewers' comments:

Reviewer #1 (Remarks to the Author):

Polycomb group genes (PcG), including Polycomb repressive complexes PRC1 and PRC2, are transcriptional repressors that regulate crucial developmental and physiological processes. PRC1 and PRC2 have also been found to play essential roles in cancer development and progression. The mechanisms by which PRC1 contribute to cancer are still poorly understood. Evidence is accruing that, besides gene silencing, PRC1 can also activate gene expression. The study by Chan and colleagues shows that PRC1 genes are amplified in breast cancer and that RING1B is recruited to enhancers to regulate oncogenic transcriptional programs. In breast cancer as well as other cancers, the authors also find that RING1B directly regulates chromatin accessibility at enhancers that regulate oncogenes.

This work is clearly of interest and the well executed experiments in this manuscript certainly merits serious consideration. Before I can recommend publication, several issues require clarifications and/or further experimentations.

Specific comments:

1. ChIPSeq experiments are performed in several parts of the manuscript. While these are of good quality and adequately analyzed, further validation by qPCR of some targets would be useful, e.g. for Fig. 1.
2. In Fig. 1B-F, ChIP-seq could be performed in RING1B knockdown cells to truly assess whether the identified regions are indeed RING1B targets.
3. In Fig. 4, it is claimed that RING1B depletion affects several key oncogenic genes, which subsequently impairs signalling pathways involved in breast cancer progression and metastasis (EMT, TGF β ,...). Previous reports (e.g. Bosch et al., *Oncotarget* 2014) showed that RING1B is required for cell migration and invasion in breast cancer cell lines. Hence, it would be nice if authors could provide functional experiments linking RING1B depletion to phenotypic characteristics in T47D and MDA-MB-231 cancer cell lines, such as proliferation or migration.
4. A recent publication highlighted a functional cross-talk RING1B and EZH2 (Hernández et al., *Carcinogenesis* 2018). Hence, authors should discuss about such recent work in light of their study.
5. RING1B amplification showed correlation to its overexpression in breast cancer compared to normal breast tissues, regardless of breast cancer subtype (Fig. S1E-F). However, authors only selected two breast cancer cell lines: T47D (ER+ luminal A) and MDA-MB-231 (Triple negative). Could an HER2+ cell line be tested?
6. In Fig. S1, authors analyzed gene amplification in a large-scale genomic dataset, showing that canonical PRC1 subunits are preferentially amplified as compared to non-canonical ones. However, additional non-canonical PRC1 subunits were not analyzed, such as KDM2B/FBXL10, WDR5. Thus, further and more detailed analyses should be performed.

Reviewer #2 (Remarks to the Author):

This study examines the gene regulatory functions of proteins in cPRC1 complex using various cancer models and iPSCs. Among several proteins in the complex, chromatin binding properties of RING1B were examined extensively in three cell lines using ChIP-seq and ATAC-seq. Authors

conclude that positive and negative gene regulation by RING1B involves altered chromatin accessibility to oncogenic transcription factors.

While the experiments are very well performed and utilized public databases extensively to justify conclusions, scientific premise for the study and results obtained are not compatible and that issue was not included in the discussion. Details are provided below.

1) Authors first utilize TCGA and other datasets to determine status of components of cPRC1 complex in various cancers and identify amplification or upregulation at transcript levels of various components. RNF2 encoding RING1B was amplified in 22% of breast cancers and is overexpressed, although not correlating with a specific subtype of breast cancer. However, RNA-seq studies with RING1B knockdown in breast cancer cell lines showed upregulation of pro-metastatic and cancer stem cell-associated genes including CD36, CD44, DUSP1, IL6, MMP1, and MMP9. If RING1B amplification/overexpression was associated with cancer progression, as predicted from public database analyses, knocking this gene would have reduced the expression of pro-metastatic genes. Thus, the assertion that RING1B reprograms chromatin to allow access to oncogenic transcription factors is not compatible with RNA-seq data. It could be that model system used does not reflect tumors in patients.

2) Concordance between RING1B-containing super-enhancers and the number of genes whose expression was affected after RING1B depletion was extremely low. For example, in T47D cells, of the 398 genes identified to contain RING1B bound super-enhancers, only 33 were deregulated upon RING1B. These results suggest that one of the two assays is not done correctly or RING1B binding has very little effect on downstream gene expression. Since candidate genes identified in the high throughput assays have not been independently verified by ChIP-assay and compatibility between results of high throughput assays and validation experiments are typically notoriously low, few of the conclusions are not well justified.

3) Based on the known function of FOXA1 as a pioneer factor and enrichment of FOXA1 binding sites on regions bound by RING1B suggest that the majority of the effects of RING1B is through FOXA1 regulation in at least T47D cells. There is need to independently verify the effect of RING1B on FOXA1 expression (not RNA-seq FPKM numbers) and to show that FOXA1-depletion affects RING1B binding.

4) Differences in RING1B binding between iPSC cells, MCF10A, T47D and MDA-MB-231 may not have any relationship to carcinogenesis but could be a reflection of cell-of-origin. In this context, MCF10A, T47D and MD-MB-231 cells showed substantial overlap, despite these cell lines corresponding to basal, luminal and mesenchymal stem-like subtypes of breast/breast cancer.

5) Several of the enrichment analyses between cell types are not analyzed statistically.

Response to Reviewer #1:

We thank the reviewer for the comment that *“This work is clearly of interest and the well executed experiments in this manuscript certainly merits serious consideration.”*

“1. ChIP-seq experiments are performed in several parts of the manuscript. While these are of good quality and adequately analyzed, further validation by qPCR of some targets would be useful, e.g. for Fig. 1.”

As suggested by the reviewer, we have now included multiple ChIP-seq validations throughout the revised version of the manuscript:

1. To further validate the RING1B ChIP-seq signal in iPSCs, we performed ChIP-qPCR of PRC1/2 targets identified in our RING1B ChIP-seq, using another RING1B antibody (Cell Signaling #5694) (Supplementary Fig. 3c).
2. We further validated the RING1B ChIP-seq signal from T47D cells by performing RING1B ChIP-qPCR of several RING1B-containing enhancers. As an additional control, we performed these experiments using a second RING1B antibody (Cell Signaling #5694) (Fig. 5b-left panel).
3. We performed BRD4 ChIP-qPCR of BRD4/RING1B-containing enhancers in MDA-MB-231 cells. As controls, we used both IgG and cells treated with the BRD4 inhibitor JQ1. As expected, BRD4 signal was decreased upon 48h of treatment with JQ1 (Supplementary Fig. 5e).
4. We also performed ChIP-qPCR for several histone modifications, including H3K27me3, H3K4me3, H3K4me1 and H3K27me3 in iPSCs and T47D cells (Supplementary Fig. 3d and Fig. 5b-right panel)

“2. In Fig. 1B-F, ChIP-seq could be performed in RING1B knockdown cells to truly assess whether the identified regions are indeed RING1B targets.”

While we agree that RING1B ChIP-seq in RING1B KD cells would be useful, we believe that we provided several lines of evidence strongly indicating that the RING1B ChIP-seq is specific. Here we briefly describe the most important experiments:

1. For the RING1B ChIP-seq experiments, we used a mass-spectrometry validated RING1B antibody from Active Motif which has been extensively used (Liu et al., 2017; Neijts et al., 2016; Yakushiji-Kaminatsui et al., 2016). We further validated, by western-blot and immunoprecipitation, the specificity of the antibody using RING1B-depleted MDA-MB-231 cells (Supplementary Fig. 3a-b).
2. RING1B target genes in iPSCs/stem cells have been extensively mapped by many laboratories including ours (Morey et al., 2013; 2012; 2015b). Therefore, we used human iPSCs as an internal control for RING1B ChIP-seq experiments. Importantly, our RING1B ChIP-seq in iPSCs is highly similar to published RING1B ChIP-seq in human ESCs or iPSCs (Lee et al., 2006). As mentioned above, we have included RING1B ChIP-qPCR assays as a further validation control (Supplementary Fig. 3c).
3. We applied very stringent bioinformatics analysis (fold change > 4 over input ChIP-seq signal, and q-value > 0.05) to assess RING1B ChIP-seq peaks.
4. The fact that RING1B binds to genomic regions and genes that are biologically relevant in each of the four cell lines indicates that the ChIP-seq signal is highly specific (Fig. 1c and 1f).
5. RING1B co-localization with H2AK119ub1 in all the cell lines used in this study also validates the RING1B ChIP-seq (Fig. 1i and Supplementary Fig. 3j-l).
6. Another PRC1 subunit, PCGF2/MEL18, was co-recruited with RING1B to a large number of genomic sites, including genes and enhancers in T47D and MDA-MB-231 (Fig. 3).
7. Moreover, we found that RING1B targets contain ESR1 binding sites in T47D cells (Fig. 2g-h), which was confirmed by our own ER α ChIP-seq (Fig. 3c-d and 3i). Similarly, analysis of RING1B binding sites in MDA-MB-231 cells, suggested that BRD4 co-occupies RING1B sites (Fig. 2h). This result was further confirmed by BRD4 ChIP-seq (Fig. 3f-g and 3j) and also in this revised version by ChIP-qPCR (Supplementary Fig. 5e).
8. We showed by ATAC-seq that the chromatin accessibility at a large number of enhancers containing RING1B sites were affected upon RING1B knockdown (Fig. 6). Importantly, most of the chromatin accessibility sites regulated by RING1B were enhancer regions co-occupied by RING1B/PCGF2/ER α in ER+ T47D cells and RING1B/PCGF2/BRD4 in MDA-MB-231 cells (Fig.

- 6). Thus, ATAC-seq, further confirmed that RING1B functionally binds to enhancers.
9. Expression of 65% of enhancer-RNA of RING1B-bound enhancers were affected upon RING1B depletion (Fig. 4e and Supplementary Fig. 6h).
10. Analyses of RING1B ChIP-seq from two ENCODE datasets indicated that RING1B binds to active enhancers in other cancer cells (Fig. 7). Moreover, other groups have observed in *Drosophila* (Loubiere et al., 2016) and melanoma cells (Rai et al., 2015) that a large number of RING1B targets contain H3K27ac and are devoid of H3K27me3.
11. During the revision of our manuscript, the laboratory of Dr. Ezhkova published a paper in *Cell Stem Cell* in which they dissected the role of RING1B/PRC1 in adult epidermal stem cells (Cohen et al., 2018). The authors showed that RING1B co-localizes with a subset of genomic regions containing active enhancer marks (Fig. 1b and 1e). This study is in full agreement with our results showing that RING1B/PRC1 can exert dual repressive and active functions.

In the revised version of the manuscript, we provide the following new experiments to further confirm the specificity of the RING1B ChIP-seq signal:

1. As mentioned in point #1, we performed ChIP-qPCR validations using a second RING1B antibody (Cell Signaling #5694) in control and RING1B KD T47D cells (Fig. 5b).
2. We performed ChIP-qPCR validations using a second RING1B antibody in iPSCs (Fig. 3c).

“3. In Fig. 4, it is claimed that RING1B depletion affects several key oncogenic genes, which subsequently impairs signalling pathways involved in breast cancer progression and metastasis (EMT, TGFB,...). Previous reports (e.g. Bosch et al., Oncotarget 2014) showed that RING1B is required for cell migration and invasion in breast cancer cell lines. Hence, it would be nice if authors could provide functional experiments linking RING1B depletion to phenotypic characteristics in T47D and MDA-MB-231 cancer cell lines, such as proliferation or migration.”

We thank the reviewer for this suggestion and we apologize for not discussing the important findings described in Bosch et al. This paper is now properly discussed in the revised version of our manuscript. To address whether RING1B is important for breast cancer tumorigenesis and/or metastasis, we performed mouse xenografts experiments. We first generated T47D and MDA-MB-231 cells expressing luciferase and GFP to monitor the growth and metastasis by IVIS (In Vivo Imaging System). Stable T47D^{luc-GFP} and MDA-MB-231^{luc-GFP} cells were then transduced either with shRNA-control or shRNA-RING1B (Supplementary Fig. 7b). After selection with puromycin, cells were injected into the mammary fat pad of NSG mice (n=5 per group). Tumor growth (*in vivo* proliferation) and metastasis (*in vivo* migration) were monitored for ~70 days.

1. Assessment of the role of RING1B in ER+ tumorigenesis: T47D cells are not highly metastatic (Holliday and Speirs, 2011), but RING1B-depleted T47D cells were more aggressive than control tumors and invaded the lungs (Fig. 4i). Although we did not observe significant changes in either primary tumor growth or weight (Supplementary Fig. 7c-top panel and Supplementary Fig. 7d-top panel), mice injected with shRNA-RING1B cells lost more weight than mice injected with shRNA-control cells (Supplementary Fig. 7e-top panel). These results are in agreement with our new analysis using TCGA data from patients with ER+ breast cancer and low/high levels of RING1B. Briefly, we found that patients with ER+ breast cancer and high levels of RING1B have a better survival prognosis than patients with low levels of RING1B (Fig. 4h-top panel). Moreover, this data is in agreement with our initial hypothesis that RING1B-depleted cells may be more aggressive than control cells, as these cells aberrantly upregulated *CD36*, the only known marker of the metastasis-initiating cancer stem cell population described to date (Pascual et al., 2017) and two lung metastasis markers (*EREG*, *ID1*) (Minn et al., 2005) (Fig. 4a-b). We further confirmed by RT-qPCR and western-blot that *CD36* was strongly upregulated in T47D-shRING1B cells (Fig. 4c and Supplementary Fig. 7a).
2. Assessment of the role of RING1B in TNBC tumorigenesis: In contrast to the function of RING1B in ER+ cells, depletion of RING1B in the highly metastatic MDA-MB-231 cells significantly reduced their invasive potential (Fig. 4j). Importantly, this result is in agreement with the paper mentioned

by the reviewer (Bosch et al., 2014) and is in full agreement with our initial hypothesis advocating for an opposite role of RING1B in ER+ and TNBC metastasis. Moreover, TCGA data from basal breast cancer patient samples indicated that high levels of RING1B expression is associated with lower survival prognosis (Fig. 4h).

From this set of experiments, we conclude that RING1B has a divergent role in breast cancer subtypes. We show that *RNF2* is a pro-metastatic gene in TNBC and an anti-metastatic gene in ER+ breast cancer cells. We hypothesize that RING1B-mediated regulation of *CD36* in ER+ is required to prevent lung metastasis. In TNBC, CD36-mediated fatty acid metabolism is strongly impaired upon RING1B depletion (Fig. 4b and Supplementary Fig. 6b). Further experiments will be required to fully understand the opposite role of RING1B in breast cancer metastasis through the CD36 and fatty acid metabolic pathway and the negative crosstalk with the EMT pathway, which is inversely affected in ER+ and TNBC cells depleted of RING1B (Figure 4a-b and Supplementary Fig. 6b).

We also performed cell cycle analysis to investigate the role of RING1B in proliferation. We arrested control and RING1B-depleted cells by serum-depletion for 72h followed by BrdU incorporation during a time-course of 24 hours. Results presented below indicate that RING1B-depleted T47D have difficulty entering the S phase upon induction into the cell cycle, as there is an accumulation of cells in G1/S at 4h, and this delay is abrogated at later time points indicating that cells can progress through the cell cycle once entered. RING1B-depleted MDA-MB-231 cells are slightly arrested in G1 phase, but to a lesser extent (Figure below). We believe that incorporation of these results may distract the reader from the main messages of this manuscript. Therefore, we would like to show this figure only in the point-by-point response to reviewers.

Figure 1. Cell cycle analysis in T47D and MDA-MB-231 cells upon RING1B knock-down. Cells were serum starved for 72h to arrest cells in the cell cycle, followed by induction with complete growth media containing 10% serum. Cells were labeled with BrdU and collected at the indicated time points, then stained with anti-BrdU and DAPI and analyzed by FACS. Both T47D and MDA-MB-231 cells exhibit a delay in cell cycle entry, with an accumulated in the G1/S transition.

“4. A recent publication highlighted a functional cross-talk RING1B and EZH2 (Hernández et al., Carcinogenesis 2018). Hence, authors should discuss about such recent work in light of their study.”

We have now included this reference in the revised manuscript highlighting that RING1B levels are significantly higher in primary cutaneous squamous cell carcinoma tumors (cSCCs) that can metastasize, suggesting that RING1B have similar functional roles in cSCCs and TNBC.

“5. RING1B amplification showed correlation to its overexpression in breast cancer compared to normal breast tissues, regardless of breast cancer subtype (Fig. S1E-F). However, authors only selected two breast cancer cell lines: T47D (ER+ luminal A) and MDA-MB- 231 (Triple negative). Could an HER2+ cell line be tested?”

We thank the reviewer for this suggestion. We have performed stable RING1B depletion in SKBR3 cells, a commonly used HER2+ cell line, and performed RNA-seq experiments (Supplementary Fig. 6d-f). With this experiment, we aimed to determine whether RING1B also regulates genes involved in tumorigenesis and metastasis in breast cancer cells other than ER+ and TNBC.

Stable RING1B depletion affected the expression of 674 genes (q-value<0.05, FC>2), being 255 upregulated and 419 downregulated (Supplementary Fig. 6e). Because a larger number of genes were downregulated, we hypothesize that in these cells, RING1B is also involved in positively regulating gene expression in HER2+ cells. Gene-set enrichment analysis (GSEA) showed that RING1B strongly deregulated pathways involved in cancer. The upregulated gene signatures included “Focal adhesion” and “TGF- β Signaling pathway”. Downregulated gene signatures included “PPAR Signaling” and “Fatty Acid metabolism” among others (Supplementary Fig. 6f). These results indicate that RING1B regulates similar set of gene signatures in HER2+ and TNBC cells that are opposite to its targets in ER+ cells.

“6. In Fig. S1, authors analyzed gene amplification in a large-scale genomic dataset, showing that canonical PRC1 subunits are preferentially amplified as compared to non-canonical ones. However, additional non-canonical PRC1 subunits were not analyzed, such as KDM2B/FBXL10, WDR5. Thus, further and more detailed analyses should be performed.”

We apologize for not including all of the non-canonical PRC1 (ncPRC1) genes in the TCGA analysis. We did not initially include them in the first version of the manuscript since some of the ncPRC1 proteins are part of other protein complexes (i.e. WDR5 is part of MLL complexes, HDAC1/2 are stable components of NuRD complexes, etc.) or because they have PRC1-independent functions (i.e. FBXL10, BCOR, etc.). Nevertheless, as suggested by the reviewer, we now included their amplification and mutation status in Supplementary Fig. 1c and 1d.

We would like to sincerely thank this reviewer for his/her insightful suggestions and comments on our manuscript. We hope this revised version will satisfy his/her concerns. For clarity purposes, the revised text and new figure panels are highlighted in red.

Response to Reviewer #2:

We thank the reviewer for the comment that “*the experiments are very well performed and utilized public databases extensively to justify conclusions,*”. This reviewer pointed out that the “*scientific premise for the study and results obtained are not compatible and that issue was not included in the discussion.*” We apologize for the lack of clarity and we strongly believe that the revised version will satisfy the reviewer.

“1) Authors first utilize TCGA and other datasets to determine status of components of cPRC1 complex in various cancers and identify amplification or upregulation at transcript levels of various components. RNF2 encoding RING1B was amplified in 22% of breast cancers and is overexpressed, although not correlating with a specific subtype of breast cancer. However, RNA-seq studies with RING1B knockdown in breast cancer cell lines showed upregulation of pro-metastatic and cancer stem cell-associated genes including CD36, CD44, DUSP1, IL6, MMP1, and MMP9. If RING1B amplification/overexpression was associated with cancer progression, as predicted from public database analyses, knocking this gene would have reduced the expression of pro-metastatic genes. Thus, the assertion that RING1B reprograms chromatin to allow access to oncogenic transcription factors is not compatible with RNA-seq data. It could be that model system used does not reflect tumors in patients.”

We apologize for the lack of clarity. We did not intend to suggest that *RNF2* amplification was solely associated with breast cancer progression, yet we agree with the reviewer that its amplification and overexpression may suggest that *RING1B* functions as an oncogene in breast cancer. This assumption warrants precaution for multiple reasons: 1) breast cancer is a heterogeneous disease and 2) our genome-wide *RING1B* experiments indicates that *RING1B* controls gene expression through multiple mechanisms, with both repressive and activating functions in distinct breast cancer subtypes. Therefore, it is difficult to associate which of the dual *RING1B* functions is more prevalent or more important.

To better elucidate the role of *RING1B* in breast cancer and to verify that our model system reflects patient data, we performed the following new analyses and experiments that we hope will help to clarify this very important point raised by the reviewer:

1. **Kaplan-Meier patient survival prognosis plots:**

- a. We performed Kaplan-Meier (KM) plots of breast cancer patients classified by the expression of the estrogen receptor and *RNF2*. As shown in Figure 4h (upper panel), high levels of *RING1B* predicted a better survival outcome in patients with ER+ tumors. This result suggests that in ER+ breast cancer, *RING1B* might be a tumor-suppressor or an anti-metastatic gene (see below).
- b. We also performed KM plots of patients with TNBC/basal breast cancer classified by their levels of *RNF2*. As shown in Figure 4h (bottom panel), high levels of *RNF2* predicted a lower survival prognosis compared with patients expressing lower levels of *RNF2*. This result suggests that patients with TNBC/basal breast cancer, *RING1B* might function as an oncogene or a pro-metastatic gene. To further corroborate this data, it was recently shown that in TNBC cells, depletion of *RING1B* reduced cellular migration *in vitro* (Bosch et al., 2014).

These analyses suggest that the role of *RING1B* in breast cancer is more complex than previously anticipated, and that *RING1B* overexpression might exert opposite roles in TNBC and ER+ breast cancer subtypes. Importantly, these results are in agreement with the opposite gene expression signatures we found in ER+ (T47D) and TNBC (MDA-MB-231) cell lines upon *RING1B* depletion and with our xenografts experiments described below.

2. **Assessment of the role of *RING1B* in breast cancer tumorigenesis and aggressiveness (metastasis) *in vivo*.** To further determine whether our cellular model system recapitulates the prognosis of patients with ER+ and basal breast cancer, we performed orthotopic xenograft tumor models in NSG mice. To this end, we first engineered luciferase-GFP expressing T47D and MDA-MB-231 cells to monitor their tumorigenic and metastatic potential using IVIS (In Vivo Optical Imaging) (Supplementary Fig. 7b). T47D^{luc-GFP} and MDA-MB-231^{luc-GFP} cells were then transduced with either control or *RING1B* shRNA lentiviruses. After selection with puromycin, cells were

injected into the mammary fat pad of NSG mice (n=5 per group). Tumor development and metastasis were monitored with IVIS for ~70 days. Here, we describe the main results (Fig. 4h-j and Supplementary Fig. 7), which are extensively discussed in the revised version of the manuscript:

- a. Role of RING1B in tumorigenesis and metastasis of ER+ cells: When compared to control T47D cells, shRING1B-T47D cells were more aggressive and metastasized to the lungs (Fig. 4l). This result is important, as T47D cells rarely metastasize and colonize the lungs (Holliday and Speirs, 2011). Also, this result is in agreement with the gene expression profile of RING1B-KD T47D cells, in which *CD36*, the only known marker of metastasis-initiating cancer stem cells (Pascual et al., 2017), and two lung metastasis markers (*EREG* and *ID1*, Minn et al., 2005) were among the most upregulated genes (Fig. 4A). In the revised version of the manuscript, we confirmed the accumulation of CD36 in RING1B-depleted cells both at the mRNA and protein level (Fig. 4c-top panel and Supplementary Fig. 7a). Additional analysis (weight of tumors and mice, and quantification of primary tumor size) are further discussed in the revised text and included in Supplementary Fig. 7c-e. Briefly, we found no changes in size and growth of control and shRING1B-derived tumors. In agreement with the increased aggressiveness of RING1B-depleted cells, mice injected with shRING1B cells lost weight more rapidly than controls (Supplementary Fig. 7e-top panel).
- b. Role of RING1B in tumorigenesis and metastasis of TNBC: As expected, based on our new TCGA analysis (Fig. 4h-bottom panel) and the gene expression profile of RING1B-depleted MDA-MB-231 cells (Fig. 4a), RING1B depletion reduced the invasive potential of this highly metastatic cell line (Fig. 4j). This result is in agreement with the paper mentioned by reviewer #1 in which depletion of RING1B in MDA-MB-231 reduced cellular invasion *in vitro* (Bosch et al., 2014). Additional analysis (weight of tumors and mice, and quantification of primary tumors size) are further discussed in the revised text and included in Supplementary Fig. 7c-d. Briefly, we found no changes in tumor size and growth of control and shRING1B-derived tumors. In agreement with the reduced aggressiveness of RING1B-depleted cells, mice injected with shRING1B cells did not lose weight as rapidly as the control mice (Supplementary Fig. 7e-bottom panel).

Overall, we demonstrated that our model system recapitulates the TCGA data presented in the original submission, which is further supported by the new analyses and experiments presented in the revised manuscript. In conclusion, we show that the functional impact of aberrant RING1B expression in breast cancer progression varies between breast cancer subtypes, and we hypothesize that RING1B might be a novel biomarker for survival prognosis depending on the breast cancer subtype.

“2) Concordance between RING1B-containing super-enhancers and the number of genes whose expression was affected after RING1B depletion was extremely low. For example, in T47D cells, of the 398 genes identified to contain RING1B bound super-enhancers, only 33 were deregulated upon RING1B. These results suggest that one of the two assays is not done correctly or RING1B binding has very little effect on downstream gene expression.”

We apologize for the lack of clarity in the text. The analysis we performed to detect the genes potentially regulated by the RING1B-containing super-enhancers (SE) was based on the assumption that a given SE would regulate the closest transcriptionally active gene. Although this type of analysis was performed in other publications, SEs do not only regulate the expression of the nearest gene (Hnisz et al., 2015; Whyte et al., 2013; Young, 2011). To investigate whether RING1B-SEs may regulate more than 33 genes, we repeated the RING1B-SE/target gene analysis using the following rationale and methodology:

1. We used available ENCODE Hi-C data from T47D cells to identify all nearby genes of RING1B bound SEs located within TADs (Topological Associated Domains). We used this approach because genes within TADs are more likely to physically interact and have similar gene expression profiles.
2. We found 2,484 genes near RING1B-SEs located within TADs.
3. We then analyzed the expression of these genes in control and RING1B-depleted cells.

4. We found that the expression of 107 genes near the 404 RING1B-SEs within TADs were deregulated upon RING1B depletion (Fig. 4f-g).
5. Thus 26% of genes putatively controlled by RING1B-SEs were transcriptionally deregulated in RING1B-depleted cells.

In agreement with our previous analysis, these 107 genes are also occupied by ER α (Fig. 4f). Furthermore, RING1B appears to have a dual role in their transcriptional regulation (Fig. 4g). These results not only confirmed our initial analysis but strengthen our conclusion that RING1B binding to SEs is required for the transcriptional regulation of putative SE-target genes.

As mentioned by the reviewer, another possibility is that RING1B has very little effect on gene expression. Although our new analysis indicates that ~26% of genes close to a RING1B-containing SE is affected upon RING1B depletion, which further confirms the functional role of RING1B in regulating SEs and genes, we would like to briefly discuss this concern raised by the reviewer. In cellular homeostasis, depletion of many epigenetic factors, including Polycomb proteins, have a limited impact on gene expression compared to depletion of transcription factors (Aloia et al., 2013; Eagen et al., 2017; Riising et al., 2014). Moreover, the number of genes deregulated upon depletion of an epigenetic factor in cellular homeostasis should not be strictly linked to the importance of the gene. One of the best examples can be found in embryonic stem cells (ESCs), in which depletion of PRC2 and PRC1 subunits only deregulate, at most, 20% of their target genes and does not affect ESC proliferation and self-renewal (Beringer et al., 2016; Morey et al., 2012; 2015b; Riising et al., 2014). However, when Polycomb-deficient ESCs are induced to differentiate, i.e., the cells are challenged, they fail to respond to differentiation cues concomitantly with a block in differentiation (Laugesen and Helin, 2014; Morey and Helin, 2010; Morey et al., 2015a).

We have now extensively discussed these analyses in the revised version of the manuscript. In conclusion, our new RING1B-SE/target gene analysis retrieved 107 potential genes regulated by RING1B-SEs which are also decorated with ER α . We thank the reviewer for his/her suggestion as it strengthened our initial results.

“Since candidate genes identified in the high throughput assays have not been independently verified by ChIP-assay and compatibility between results of high throughput assays and validation experiments are typically notoriously low, few of the conclusions are not well justified.”

We believe that we validated our high throughput assays by multiple independent experiments:

1. RING1B target genes in iPSCs/stem cells have been extensively mapped by many laboratories (Ku et al., 2008; Lee et al., 2006; Morey et al., 2012). Therefore, we used human iPSCs as an internal control for the RING1B ChIP-seq experiments in cancer cells. Importantly, our RING1B ChIP-seq in iPSCs is highly similar to published RING1B ChIP-seq in human ESCs or iPSCs (Lee et al., 2006). Additionally, we have now performed ChIP-qPCR assays using another RING1B antibody (Supplementary Fig. 3c).
2. Another PRC1 subunit, PCGF2/MEL18, was co-recruited with RING1B to a large number of genomic sites including genes and enhancers in T47D and MDA-MB-231 (Fig. 3). This result also strongly indicates that our RING1B ChIP-seq signal is specific, and therefore can be used as a validation of the RING1B ChIP-seq.
3. Moreover, we found that RING1B targets contain ER binding sites in T47D cells (Fig. 2g-h), which was confirmed by our own ER α ChIP-seq (Fig. 3c-d and 3i). Similarly, analysis of RING1B binding sites in MDA-MB-231 cells suggested that BRD4 co-occupied RING1B sites (Fig. 2h). This result was further confirmed by BRD4 ChIP-seq (Fig. 3f-g and 3j) and in the revised version of the manuscript by ChIP-qPCR (Supplementary Fig. 5e).
4. We showed by ATAC-seq that the chromatin accessibility at a large number of enhancers containing RING1B sites were affected upon RING1B knockdown (Fig. 6). Importantly, most of the chromatin accessibility sites regulated by RING1B were indeed enhancer regions co-occupied by RING1B/PCGF2/ER α in ER $^+$ and RING1B/PCGF2/BRD4 in TNBC (Fig. 6). Thus, ATAC-seq further confirmed that RING1B functionally binds to enhancers.
5. Expression of enhancer-RNA of RING1B-bound enhancers were affected upon RING1B

- depletion (Fig. 4d-e and Supplementary Fig. 6f).
- Analyses of RING1B ChIP-seq from two ENCODE RING1B datasets indicated that RING1B binds to active enhancers in other cancer cells (Fig. 7 and Supplementary Fig. 9). Other colleagues observed in *Drosophila* (Loubiere et al., 2016) and in melanoma cells (Rai et al., 2015) that a large number of RING1B targets contain H3K27ac and are devoid of H3K27me3.
 - During the revision of our manuscript, the laboratory of Dr. Ezhkova published a paper in *Cell Stem Cell* in which they dissected the role of RING1B/PRC1 in adult epidermal stem cells (Cohen et al., 2018). The authors showed that RING1B co-localizes with a subset of genomic regions containing RING1B and active enhancer marks (Fig. 1b and 1e). This study is in full agreement with our results showing that RING1B/PRC1 can exert dual repressive and active functions in adult cells and cancer cells.

We have now included additional ChIP-seq validations throughout the revised version of the manuscript:

- To further validate the RING1B ChIP-seq signal in iPSCs, we performed ChIP-qPCR of PRC1/2 targets identified in our RING1B ChIP-seq, using another RING1B antibody (Cell Signaling #5694) (Supplementary Fig. 3c).
- We further validated the RING1B ChIP-seq signal from T47D cells by performing RING1B ChIP-qPCR of several RING1B-containing enhancers in shCTR and shRING1B cells. As an additional control, we performed these experiments using a second RING1B antibody (Fig. 5b-left panel).
- We performed BRD4 ChIP-qPCR of BRD4/RING1B-containing enhancers. As controls we used both an IgG antibody and cells treated with the BRD4 inhibitor, JQ1. As expected, BRD4 signal was decreased upon 48h of treatment with JQ1 (Supplementary Fig. 5e).
- We performed ChIP-qPCR for several histone modifications, including H3K27me3, H3K4me3 and H3K27me3 in iPSC and T47D cells (Supplementary Fig. 3d, Fig. 5b-right panel)

We also validated by RT-qPCR the RNA-seq results in RING1B knockdown T47D and MDA-MB-231 cells (Fig. 4c). Additionally, we validated the aberrant upregulation of CD36 by RT-qPCR and western-blot (Fig. 4c and Supplementary Fig. 7a). The attenuated response to estrogen in shRING1B-T47D previously shown by GSEA (Fig. 4b) has also been functionally validated (Fig. 5g). We hope that this new set of experiments will be satisfactory for the reviewer.

“3) Based on the known function of FOXA1 as a pioneer factor and enrichment of FOXA1 binding sites on regions bound by RING1B suggest that the majority of the effects of RING1B is through FOXA1 regulation in at least T47D cells. There is need to independently verify the effect of RING1B on FOXA1 expression (not RNA-seq FPKM numbers) and to show that FOXA1-depletion affects RING1B binding.”

This is a very interesting suggestion. We have extensively worked to better understand the crosstalk between RING1B and FOXA1 in T47D cells (MDA-MB-231 cells do not express FOXA1). In the revised manuscript, we generated an entirely new Figure (Figure 5) describing a novel crosstalk between RING1B, FOXA1 and ER α .

In the previous version of the manuscript, we showed that RING1B and PRC2 were recruited to FOXA1 to negatively regulate its expression in MDA-MB-231 cells. In contrast, RING1B was recruited to a putative FOXA1 enhancer in T47D cells (now Fig. 5a). This result indicated that while RING1B exerts its canonical function by repressing FOXA1 in MDA-MB-231, RING1B positively regulates FOXA1 in T47D through its binding to a putative FOXA1 enhancer. As mentioned by the reviewer, FOXA1 binding sites are enriched in our ChIP-seq and ATAC-seq assays, suggesting a functional crosstalk between RING1B and FOXA1.

To better understand this transcriptional regulatory crosstalk, we first assessed whether acute depletion of RING1B, by siRNA, affected FOXA1 protein levels in T47D. In Figure 5c (left panel), we now show that FOXA1 protein was downregulated ~50% 72h after RING1B depletion. Although, FOXA1 levels returned to normal levels about two weeks after stable RING1B depletion (Fig. 5c-right panel), cellular fractionation experiments showed that both FOXA1 and ER α were displaced from chromatin and accumulated in the soluble nuclear fraction (Fig. 5d). We further confirmed that RING1B-depleted T47D cells cultured in hormone-deprived serum for 72h do not efficiently respond to estrogen (Fig. 5g). These experiments

describe a novel mechanism that can explain why RING1B-depleted cells do not efficiently respond to estrogen. Importantly, these results are in agreement with the RNA-seq data in T47D, in which one of the most downregulated pathways was the Estrogen Receptor signaling pathway (Fig. 4b).

As suggested by the reviewer, we also depleted FOXA1 to further investigate the RING1B-FOXA1 axis. To this end, we used both siRNAs and shRNAs to deplete FOXA1 in T47D cells. Both acute and stable FOXA1 depletion downregulated RING1B levels (Fig. 5e). We also performed cellular fractionation experiments in control and FOXA1-KD cells. As shown in Figure 5f, RING1B was almost fully displaced from chromatin upon FOXA1 depletion. This result may suggest that FOXA1 directly regulates RING1B expression. We did not detect FOXA1 binding to the RING1B promoter by ChIP-seq (data not shown).

Overall, our new results indicated that:

1. RING1B directly regulates *FOXA1* expression in T47D possibly by binding to a putative *FOXA1* enhancer.
2. Stable RING1B depletion displaces FOXA1 and ER α from chromatin.
3. Stable RING1B depletion leads to an accumulation of FOXA1 and ER α in the soluble nuclei fraction.
4. RING1B-depleted T47D cells do not efficiently respond to estrogen stimulation.
5. FOXA1 depletion induces degradation of RING1B, which results in reduced RING1B binding at chromatin.

Altogether, these new experiments are in agreement with the ATAC-seq, ChIP-seq and RNA-seq experiments presented in the previous version of the manuscript, which suggested that RING1B was required for the estrogen response and that there is a functional crosstalk between RING1B, FOXA1 and ER α . With these new experiments, we can further conclude that RING1B is an essential regulator of the estrogen-mediated cellular response by multiple mechanisms, including the direct regulation of *FOXA1* expression and by allowing FOXA1, and thus ER α , to bind to chromatin.

“4) Differences in RING1B binding between iPSC cells, MCF10A, T47D and MDA-MB-231 may not have any relationship to carcinogenesis but could be a reflection of cell-of-origin. In this context, MCF10A, T47D and MD-MB-231 cells showed substantial overlap, despite these cell lines corresponding to basal, luminal and mesenchymal stem-like subtypes of breast/breast cancer.”

We thank the reviewer for this comment. We believe that one of the strengths of our study is the genome-wide mapping and characterization of RING1B genomic localization in breast cancer cells and in non-tumorigenic epithelial mammary cells. We fully agree with the reviewer that there is a significant overlap of RING1B targets in epithelial differentiated and breast cancer cells. As mentioned above, during the preparation of the revised manuscript, a paper by the Ezhkova group showed that in mouse epidermal stem cells, RING1B also has canonical (gene repression) and non-canonical (gene activation) roles (Cohen et al., 2018). In this paper, the authors showed that RING1B positively regulated epidermal stem cells genes and that RING1B was recruited to genomic sites containing histone modifications associated with active enhancers. Their results implicating RING1B in both gene activation and repression are in agreement with our results and confirm that the role of RING1B in adult stem cells, cancer cells, and fully differentiated cells is much more complex than previously anticipated.

The dual function of RING1B may be exacerbated during oncogenesis resulting in accumulation of RING1B to enhancer regions that control oncogenic pathways. Here, we show that RING1B is recruited to cell-type specific genes and oncogenic super-enhancers. Thus, although RING1B acquires non-canonical functions in differentiated cells (MCF10A), in cancer, it regulates cell-type specific super-enhancers involved in tumorigenesis. In the future, it will be very interesting to systematically assess whether the pattern of RING1B occupancy in other differentiated cells is similar to transformed cells. Our observation that RING1B is also recruited to SEs that control essential genes in leukemic and hepatocellular carcinoma cells (now Figure 7 and Supplementary Fig. 9) suggest that occupancy of RING1B in oncogenic SEs is not restricted to epithelial cells. In turn, we discuss a possible general mechanism of RING1B in controlling oncogenic pathways in multiple cancer types by activating and repressive mechanisms. We extensively discussed this

idea in the revised version of the manuscript.

“5) Several of the enrichment analyses between cell types are not analyzed statistically.”

We apologize for the lack of clarity. We have now included in the figure captions the statistical method used for each of the analyses performed and the significance (p-value) for all the corresponding panels. To further illustrate the degree of significance of specific comparisons, we also performed new enrichment analyses of the specific RING1B ChIP-seq signals in each of the four cell lines (Fig. 1f and Supplemental Fig. 3g) and statistical analysis of the RING1B ChIP-seq signals at the specific super-enhancers containing RING1B in MCF10A, MDA-MB-231 and T47D cells (Fig. 2f and Supplemental Fig. 4e).

We would like to sincerely thank this reviewer for his/her insightful suggestions and comments on our manuscript. We hope this revised version will satisfy his/her concerns. For clarity purposes, the revised text and new figure panels are highlighted in red.

References

- Aloia, L., Di Stefano, B., and Di Croce, L. (2013). Polycomb complexes in stem cells and embryonic development. *Development* *140*, 2525–2534.
- Beringer, M., Pisano, P., Di Carlo, V., Blanco, E., Chammas, P., Vizán, P., Gutierrez, A., Aranda, S., Payer, B., Wierer, M., et al. (2016). EPOP Functionally Links Elongin and Polycomb in Pluripotent Stem Cells. *Molecular Cell* *64*, 645–658.
- Bosch, A., Panoutsopoulou, K., Corominas, J.M., Gimeno, R., Moreno-Bueno, G., Martín-Caballero, J., Morales, S., Lobato, T., Martínez-Romero, C., Farias, E.F., et al. (2014). The Polycomb group protein RING1B is overexpressed in ductal breast carcinoma and is required to sustain FAK steady state levels in breast cancer epithelial cells. *Oncotarget* *5*, 2065–2076.
- Cohen, I., Zhao, D., Bar, C., Valdes, V.J., Dauber-Decker, K.L., Nguyen, M.B., Nakayama, M., Rendl, M., Bickmore, W.A., Koseki, H., et al. (2018). PRC1 Fine-tunes Gene Repression and Activation to Safeguard Skin Development and Stem Cell Specification. *Stem Cell* *22*, 726–739.e727.
- Eagen, K.P., Lieberman Aiden, E., and Kornberg, R.D. (2017). Polycomb-Mediated Chromatin Loops Revealed by a Sub-Kilobase Resolution Chromatin Interaction Map. 1–53.
- Hnisz, D., Schuijers, J., Lin, C.Y., Weintraub, A.S., Abraham, B.J., Lee, T.I., Bradner, J.E., and Young, R.A. (2015). Convergence of Developmental and Oncogenic Signaling Pathways at Transcriptional Super-Enhancers. *Molecular Cell* *58*, 362–370.
- Holliday, D.L., and Speirs, V. (2011). Choosing the right cell line for breast cancer research. *Breast Cancer Res.* *13*, 215.
- Ku, M., Koche, R.P., Rheinbay, E., Mendenhall, E.M., Endoh, M., Mikkelsen, T.S., Presser, A., Nusbaum, C., Xie, X., Chi, A.S., et al. (2008). Genomewide Analysis of PRC1 and PRC2 Occupancy Identifies Two Classes of Bivalent Domains. *PLoS Genet* *4*, e1000242–14.
- Laugesen, A., and Helin, K. (2014). Chromatin repressive complexes in stem cells, development, and cancer. *Cell Stem Cell* *14*, 735–751.
- Lee, T.I., Jenner, R.G., Boyer, L.A., Guenther, M.G., Levine, S.S., Kumar, R.M., Chevalier, B., Johnstone, S.E., Cole, M.F., Isono, K.-I., et al. (2006). Control of Developmental Regulators by Polycomb in Human Embryonic Stem Cells. *Cell* *125*, 301–313.

- Liu, P.-P., Tang, G.-B., Xu, Y.-J., Zeng, Y.-Q., Zhang, S.-F., Du, H.-Z., Teng, Z.-Q., and Liu, C.-M. (2017). MiR-203 Interplays with Polycomb Repressive Complexes to Regulate the Proliferation of Neural Stem/Progenitor Cells. *Stem Cell Reports* 9, 190–202.
- Loubiere, V., Delest, A., Thomas, A., Bonev, B., Schuettengruber, B., Sati, S., Martinez, A.-M., and Cavalli, G. (2016). Coordinate redeployment of PRC1 proteins suppresses tumor formation during *Drosophila* development. *Nat Genet* 48, 1436–1442.
- Minn, A.J., Gupta, G.P., Siegel, P.M., Bos, P.D., Shu, W., Giri, D.D., Viale, A., Olshen, A.B., Gerald, W.L., and Massagué, J. (2005). Genes that mediate breast cancer metastasis to lung. *Nature* 436, 518–524.
- Morey, L., and Helin, K. (2010). Polycomb group protein-mediated repression of transcription. *Trends in Biochemical Sciences* 35, 323–332.
- Morey, L., Aloia, L., Cozzuto, L., Benitah, S.A., and Di Croce, L. (2013). RYBP and Cbx7 define specific biological functions of polycomb complexes in mouse embryonic stem cells. *CellReports* 3, 60–69.
- Morey, L., Pascual, G., Cozzuto, L., Roma, G., Wutz, A., Benitah, S.A., and Di Croce, L. (2012). Nonoverlapping functions of the Polycomb group Cbx family of proteins in embryonic stem cells. *Cell Stem Cell* 10, 47–62.
- Morey, L., Santanach, A., and Di Croce, L. (2015a). Pluripotency and Epigenetic Factors in Mouse Embryonic Stem Cell Fate Regulation. *Molecular and Cellular Biology* 35, 2716–2728.
- Morey, L., Santanach, A., Blanco, E., Aloia, L., Nora, E.P., Bruneau, B.G., and Di Croce, L. (2015b). Polycomb Regulates Mesoderm Cell Fate-Specification in Embryonic Stem Cells through Activation and Repression Mechanisms. *Cell Stem Cell* 17, 300–315.
- Neijts, R., Amin, S., van Rooijen, C., Tan, S., Creyghton, M.P., de Laat, W., and Deschamps, J. (2016). Polarized regulatory landscape and Wnt responsiveness underlie Hox activation in embryos. *Genes Dev.* 30, 1937–1942.
- Pascual, G., Avgustinova, A., Mejetta, S., Martín, M., Castellanos, A., Attolini, C.S.-O., Berenguer, A., Prats, N., Toll, A., Hueto, J.A., et al. (2017). Targeting metastasis-initiating cells through the fatty acid receptor CD36. *Nature* 541, 41–45.
- Rai, K., Akdemir, K.C., Kwong, L.N., Fiziev, P., Wu, C.J., Keung, E.Z., Sharma, S., Samant, N.S., Williams, M., Axelrad, J.B., et al. (2015). Dual Roles of RNF2 in Melanoma Progression. *Cancer Discovery* 5, 1314–1327.
- Riising, E.M., Comet, I., Leblanc, B., Wu, X., Johansen, J.V., and Helin, K. (2014). Gene silencing triggers polycomb repressive complex 2 recruitment to CpG islands genome wide. *Molecular Cell* 55, 347–360.
- Whyte, W.A., Orlando, D.A., Hnisz, D., Abraham, B.J., Lin, C.Y., Kagey, M.H., Rahl, P.B., Lee, T.I., and Young, R.A. (2013). Master Transcription Factors and Mediator Establish Super-Enhancers at Key Cell Identity Genes. *Cell* 153, 307–319.
- Yakushiji-Kaminatsui, N., Kondo, T., Endo, T.A., Koseki, Y., Kondo, K., Ohara, O., Vidal, M., and Koseki, H. (2016). RING1 proteins contribute to early proximal-distal specification of the forelimb bud by restricting Meis2 expression. *Development* 143, 276–285.
- Young, R.A. (2011). Control of the Embryonic Stem Cell State. *Cell* 144, 940–954.

REVIEWERS' COMMENTS:

Reviewer #1 (Remarks to the Author):

The authors have put a commendable effort in answering all the points raised in the original version of the manuscript. The current version is indeed clearly improved and there is a more consistent and logical flow of the manuscript. Several sets of new data were included following suggestions to support their interpretations and conclusions. Below are my specific comments with minor suggestions:

1. The novel ChIP-qPCR results are satisfactory. However, the authors could perhaps elaborate in the main text on the criteria used for selection the representative genes for their experiments. For example, in Supplementary Fig. 3c, what criteria were used for selection of Hand1 and TBX20? Additionally, the authors could also highlight overall agreement between ChIP-seq signal and ChIP-qPCR enrichment values.
2. The explanation and the experiments used to address my initial comment are satisfactory.
3. The in vivo experiments added in light of answering my point are impressive and add a lot of value to improve the overall quality of the manuscript. However, their conclusion that RING1B has completely opposite roles in breast cancer subtypes might also be due to the existence of another potential isoform of RING1B (Ota et al. Nature Genetics 2004.). Hence, authors should add a brief discussion about such possibility, unless they could rule it out via a simple RT-qPCR.
4. The answer to this point is satisfactory.
5. The experiments done on SKBR3 cells are adequate as they clearly highlight the regulatory role of RING1B in breast cancer cells other than ER+ and TNBC.
6. The answer to this point is convincing.

Reviewer #2 (Remarks to the Author):

Authors have addressed most of the concerns raised. Differential effects of RNF2 on survival based on ER status is interesting and can potentially take the field in new direction.

Response to Reviewer #1

We thank the reviewer for their comment “*The authors have put a commendable effort in answering all the points raised*” and that “*The current version is indeed clearly improved*”.

“1. *The novel ChIP-qPCR results are satisfactory. However, the authors could perhaps elaborate in the main text on the criteria used for selection the representative genes for their experiments. For example, in Supplementary Fig. 3c, what criteria were used for selection of Hand1 and TBX20? Additionally, the authors could also highlight overall agreement between ChIP-seq signal and ChIP-qPCR enrichment values.*”

We thank the reviewer for this suggestion and apologize for not being more clear in justifying the selection of genes in Supplementary Fig. 3c. We did mention that these genes were “known RING1B target genes in iPSCs” but did not include a reference for further clarification. A reference has been added to support the selection of these genes. Additionally, we have now added a statement (page 6, Results section, *RING1B binding is redistributed in breast cancer cells*, end of paragraph 1) confirming the agreement between ChIP-seq signal and ChIP-qPCR enrichment values.

“2. *The explanation and the experiments used to address my initial comment are satisfactory.*”

We thank the reviewer again for his/her initial suggestion.

“3. *The in vivo experiments added in light of answering my point are impressive and add a lot of value to improve the overall quality of the manuscript. However, their conclusion that RING1B has completely opposite roles in breast cancer subtypes might also be due to the existence of another potential isoform of RING1B (Ota et al. Nature Genetics 2004.). Hence, authors should add a brief discussion about such possibility, unless they could rule it out via a simple RT-qPCR.*”

We thank the reviewer for this additional insight. To rule out the possibility that the novel RING1B function we describe is not due to a potential RING1B isoform, we used the ENSEMBL gene track in the UCSC genome browser, which includes all annotated transcript variants per gene. We loaded RNA-seq of T47D shCTR and shRING1B, as well as ChIP-seq signals of H3K4me3 and H3K36me3. In the genome browser screenshot below we show that potentially, 4 isoforms of *RNF2* exist:

231 the same RING1B full length isoform is produced.

“4. *The answer to this point is satisfactory.*”

We thank the reviewer again for his/her initial suggestion.

“5. *The experiments done on SKBR3 cells are adequate as they clearly highlight the regulatory role of RING1B in breast cancer cells other than ER+ and TNBC.*”

We thank the reviewer again for his/her suggestion.

“6. *The answer to this point is convincing.*”

We thank the reviewer again for his/her suggestion.

Response to Reviewer #2

We strongly agree with the reviewer that *“Differential effects of RNF2 on survival based on ER status is interesting and can potentially take the field in new direction”*.

We thank this reviewer again for his/her suggestions.